# PTAD: Prototype-Oriented Tabular Anomaly Detection via Mask Modeling

## Abstract

Tabular anomaly detection, which aims at identifying deviant samples, has been crucial in a variety of real-world applications, such as medical disease identification, financial fraud detection, intrusion monitoring, etc. Although recent deep learning-based methods have achieved competitive performances, these methods suffer from representation entanglement and the lack of global correlation modeling, which leads to the 'abnormal leakage' issue and hinders anomaly detection performance. To tackle the problem, we incorporate mask modeling and prototype learning into tabular anomaly detection. The core idea is to design learnable masks by disentangled representation learning within a projection space and extracting normal dependencies as explicit global prototypes. Specifically, the overall model involves two parts: (i) During encoding, we perform mask modeling in both the data space and projection space with orthogonal basis vectors for masking out the suspicious abnormal locations; (ii) During decoding, we decode multiple masked representations in parallel for reconstruction and learn association prototypes to extract normal characteristic correlations. Our proposal derives from a distribution-matching perspective, where both projection space learning and association prototype learning are formulated as optimal transport problems, and the calibration distances are utilized to refine the anomaly scores. By conducting both quantitative and qualitative experiments on 20 tabular benchmarks, our model surpasses other competitors and possesses good interpretability.

## 1 Introduction

Tabular data, often structured as tables in relational databases with rows signifying individual data samples and columns representing feature variables, have become indispensable across diverse real-world domains including healthcare (Hernandez et al., 2022), engineering (Ye et al., 2023), finance (Assefa et al., 2020), etc. Tabular anomaly detection (AD), which endeavors to identify samples that diverge from a pre-defined notion of normality, plays a pivotal role in diverse scientific and industrial contexts, such as medical disease identification (Fernando et al., 2021), financial fraud detection (Al-Hashedi & Magalingam, 2021), cybersecurity intrusion monitoring (Malaiya et al., 2019), and astronomy (Reyes & Estévez, 2020). In practical scenarios, obtaining labeled anomalies is always impractical or prohibitive, necessitating a common implementation of training solely on normal samples. By distilling the inherent characteristic patterns from normal training data, anomalies are expected to be detected with deviations from normal patterns (Ruff et al., 2021). Nevertheless, the intricate, heterogeneous, and unstructured nature of tabular data features (Chang et al., 2023) poses significant challenges in identifying such characteristic patterns.

Recent works (Qiu et al., 2021; Shenkar & Wolf, 2022) have highlighted the importance of considering the particular characteristics of tabular data. For example, Neutral AD (Qiu et al., 2021) and ICL (Shenkar & Wolf, 2022) employ contrastive learning-based loss functions to create pretext tasks for tabular data, where the characteristic patterns are modeled by the contrastive losses and samples with a high loss value indicate a high possibility of anomaly. Recently, several models adhere to the reconstruction pipeline to capture characteristic patterns during reconstruction, which achieves state-of-the-art (SOTA) performances for tabular anomaly detection. In particular, NPT-AD (Thimonier et al., 2024) leverages Non-Parametric Transformers (NPT) (Kossen et al., 2021) to capture both feature-feature and sample-sample dependencies for anomaly detection during reconstructing tabular data. MCM (Yin et al., 2024) designs a learnable masking strategy to capture

intrinsic correlations between features in training data and detect anomalies with reconstruction errors. Typically, the motivation behind these methodologies is that a well-trained model struggles to generate or represent samples that deviate significantly from the normal distribution (Yin et al., 2024; Pang et al., 2021). Nevertheless, reconstruction-based AD methods may fall into an 'anomaly leakage' issue, where both normal and anomalous samples can be well recovered, and hence fail to detect outliers (You et al., 2022; Li et al., 2024). The inherent reasons might reside in the representation entanglement, where different features or relations of the learned data representations are highly correlated and entangled with each other, impeding anomaly discriminability and precise anomaly detection. Furthermore, the representations tend to overlook global correlation patterns as each data sample is represented distinctively, which fails to model the shared normal information among distinct normal samples and thus hinders the detection performance (Ye et al., 2024).

To tackle the above issues, we introduce PTAD, a prototype-oriented tabular anomaly detection method for tabular AD. The fundamental concepts center on two key aspects: 1) To prevent leaking anomaly information, we propose data-adaptive masking strategies to find suspicious anomaly locations and reconstruct them with normal information, thus resulting in large deviations for instructing anomalies. Specifically, a data-space soft masking strategy and a projection-space multiple masking strategy are designed to select optimal masks. Furthermore, to encourage disentangled representation learning, projection space is constructed based on a group of learnable orthogonal basis vectors. Furthermore, to capture various data characteristics and diverse inherent relationships, we introduce a multiple mask strategy in projection space while saving computational consumption. 2) Considering the characteristics of tabular data are heterogeneous and complex, we investigate the correlation patterns between features to facilitate modeling of tabular normal patterns and detecting anomalies, termed association prototype learning. The processes of basis vectors learning and association prototypes learning are formulated as optimal transport (OT) problems from a distribution-matching perspective, in which the transport cost can naturally serve as a criterion for anomaly assessment as it detects the deviation degree from the learned normality patterns.

In brief, our main contributions are summarized as follows: (1) We introduce a novel mask modeling method for relieving the leakage of anomalies, which aligns the distribution over p-space representations and the distribution over disentangled orthogonal basis vectors. (2) We investigate the learning of global correlation patterns in tabular data via solving an OT problem and explore a novel direction of incorporating association prototypes for tabular AD. (3) Extensive experiments on various datasets demonstrate the superiority and interpretability of our method for tabular AD.

## 2 RELATED WORK

**Tabular Anomaly Detection.** Over the past decades, numerous methods for tabular AD have been developed to identify significant deviations from the majority of data objects, which can be roughly divided into four groups: ***i) Supervised methods.*** With the availability of both normal and abnormal training samples, supervised methods such as Support Vector Machine (SVM) (Hearst et al., 1998) and deep networks (Gorishniy et al., 2021) developed, however, facing the risk of missing unknown anomalies. ***ii) Semi-supervised methods.*** Capitalizing the supervision from partial labels, the semi-supervised algorithms (Villa-Pérez et al., 2021; Pang et al., 2023) efficiently use the partially labeled data and facilitate representation learning with the unlabeled data. ***iii) Unsupervised methods.*** Without any label information of training data, unsupervised methods aim to find deviations from the majority of data, e.g. deep autoencoders (Kim et al., 2019; Han et al., 2022) and GANs (Schlegl et al., 2017; Sabuhi et al., 2021) suppose abnormality can be indicated with high reconstruction error. ***iv) Self-Supervised method.*** Several recent studies have revealed that self-supervised learning facilitates anomaly detection by creating pretext tasks to train neural networks for modeling better characteristics within training data. In particular, NPT-AD (Thimonier et al., 2024) leverages Non-Parametric Transformers for anomaly detection to capture both feature-feature and sample-sample dependencies while reconstructing tabular data. Additionally, MCM (Yin et al., 2024) extends the mask modeling to tabular AD, which generates diverse multiple masks and jointly utilizes its reconstructions for anomaly detection. However, reconstruction models usually suffer from the 'anomaly leakage' issue, and the reconstruction error as a general anomaly detection score, is limited for clear and precise anomaly detection. This motivates us to perform mask modeling and prototype learning to relieve anomaly reconstruction and find a new indicator for anomaly scoring.

**Prototype Learning.** Prototype learning has been widely studied in different tasks of computer vision (Nauta et al., 2021; Zhou et al., 2022), and natural language processing (Huang et al., 2012; Zalmout & Li, 2022). Typically, prototypes refer to empirical proxies and are computed as the weighted results of latent features of all instances of a particular class, and the distances to prototypes facilitate classification, recognition, representations, etc. Recently, prototype learning has been introduced to image anomaly detection to facilitate extracting normal feature representations to distinguish anomalous samples. In particular, HVQ-Trans (Lu et al., 2023) preserves the typical normal features as discrete iconic prototypes for image reconstruction via vector quantization. Furthermore, VPDM (Li et al., 2024) leverages prototypes as vague information about the target into a conditional diffusion model to incrementally enhance details for reconstruction. However, tabular data exhibits heterogeneous, intricate features devoid of a rigid structure (Chang et al., 2023), posing significant challenges in identifying distinctive characteristic patterns. Simply adopting a straightforward approach to extract feature prototypes is inadequate for tabular data. Consequently, we are motivated to learn the intricate correlation patterns among features, termed association prototypes, rather than focusing solely on the features themselves, to enhance the capabilities of tabular AD.

## 3 PRELIMINARY

**Problem Formulation.** This paper aims at tabular AD, where the training set only contains normal samples following the one-class classification setting. Denoting the training set of $N_{train}$ in-class normal samples as $D_{\text{train}} = \{\mathbf{x}_n\}_{n=1}^{N_{train}}$, where each sample is a $d$-dimensional vector. Denoting the testing set of $N_{test}$ samples as $D_{\text{test}} = \{\mathbf{x}_n\}_{n=1}^{N_{test}}$, which contains both normal and abnormal samples. The objective of tabular AD is to develop an anomaly scoring function $\mathcal{S} : \mathbb{R}^d \to \mathbb{R}$ that assigns low scores to samples drawn from the same underlying distribution as $D_{\text{train}}$ and high scores to the samples not aligned with $D_{\text{train}}$. Typically, standard reconstruction-based approaches (Yin et al., 2024; Thimonier et al., 2024) learn a mapping function $\mathbf{\Phi}_\theta : \mathbb{R}^d \longrightarrow \mathbb{R}^d$ by minimizing the reconstruction loss, which is often employed as the measurement of anomaly score.

**Non-Parametric Transformer.** Non-Parametric Transformers (NPTs) have shown the priority of reasoning about relationships between both datapoints and features (Kossen et al., 2021; Thimonier et al., 2024) for tabular data. Specifically, each NPT layer involves an attention between datapoints (ABD) layer and an attention between attributes (ABA) layer to capture sample-sample and feature-feature dependencies, respectively. NPT receives the data ($\mathbf{X} \in \mathbb{R}^{N \times d}$ and stochastic masking matrix with the same dimention as input, then maps them through a linear mapping into $\mathbf{H}^0 \in \mathbb{R}^{N \times d \times e}$ by transforming each feature of each sample in data space into an e-dimensional embedding. Next, NPT applies ABD and ABA alternatively. For the $l^{th}$ ABD layer, the embedding is flattened to $\mathbb{R}^{N \times H}$ with $H = d \times e$, and then multi-head self-attention (MHSA) is applied across all samples. For the $l^{th}$ ABA layer, we reshape the embedding as $\mathbb{R}^{N \times d \times e}$ and then apply MHSA independently to each row (i.e. a single datapoint) across the feature dimension. The ABD and ABA layer can be formulated as follows:

$$\text{ABD}(\mathbf{H}^l) = \text{MHSA}(\mathbf{H}^l) = \mathbf{H}^{l+1} \in \mathbb{R}^{N \times H},$$

$$\text{ABA}(\mathbf{H}^l) = \underset{\text{axis}=N}{\text{Stack}} \left( \text{MHSA}(\mathbf{H}_1^l), ..., \text{MHSA}(\mathbf{H}_N^l) \right) = \mathbf{H}^{l+1} \in \mathbb{R}^{N \times H}. \quad (1)$$

By alternatively conducting ABD and ABA, NPT is trained to reconstruct the stochastic masked input and model intrinsic dependencies among datapoints and within each datapoint. Motivated by NPT (Kossen et al., 2021), Thimonier et al. (2024) introduce NPT-AD by incorporating both sample-sample and feature-feature dependencies in tabular AD, which showcases its effectiveness and superiority for tabular data. However, it needs to combine the validation samples and the entire training set for detecting samples during inference, which results in large computation costs and potentially compromising applicability to big datasets.

**Optimal Transport.** OT has a rich theoretical foundation (Dvurechensky et al., 2018; Chizat et al., 2018; Courty et al., 2016), which measures the minimal cost to transport between two probability distributions. Here, we only focus our discussion on OT for discrete probability distributions and please refer to Peyré et al. (2019) for more details. Denote two discrete probability distributions over an arbitrary space $S \in \mathbb{R}^d$ as $p = \sum_{i=1}^n a_i \delta_{x_i}$ and $q = \sum_{j=1}^m b_j \delta_{y_j}$, where both $\boldsymbol{a} \in \sum^n$ and $\boldsymbol{b} \in \sum^m$ are discrete probabilities summing to 1. The OT distance between $p$ and $q$ is defined as

$$\text{OT}(p, q) = \min_{\mathbf{T} \in \Pi(p,q)} \langle \mathbf{T}, \mathbf{C} \rangle, \quad (2)$$

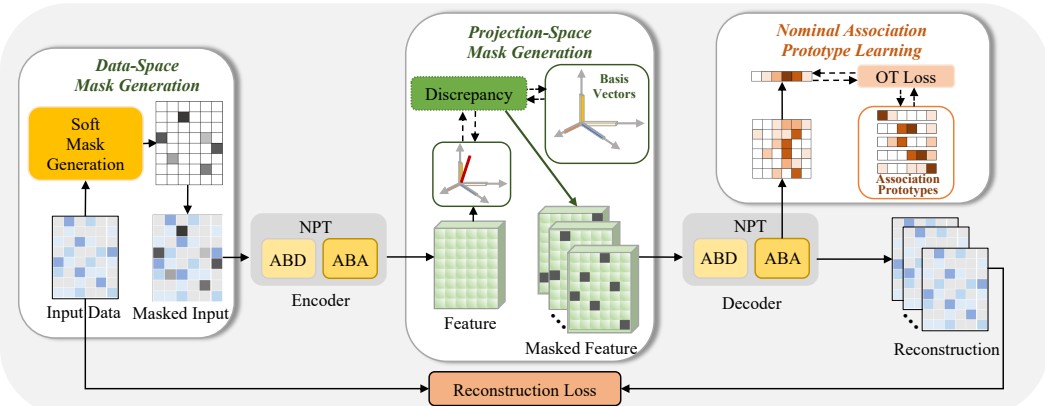

Figure 1: Overall framework: The input data is softly masked in data space and encoded through an NPT layer; Then, multiple masks are generated via discrepancies with P-space basis vectors; Multiple masked features are decoded by an NPT layer, during which the association prototypes are learned for instructing anomalies.

where $\langle \cdot, \cdot \rangle$ is the Frobenius dot-product and $\mathbf{C} \in \mathbb{R}^{n \times m}_{\geq 0}$ is the transport cost matrix where $C_{ij} = \text{Distance}(x_i, y_j)$ reflects the cost between $x_i$ and $y_j$. The transport probability matrix $\mathbf{T} \in \mathbb{R}^{n \times m}_{\geq 0}$ is subject to $\Pi(p, q) := \{\mathbf{T} | \sum_{i=1}^{n} T_{ij} = b_j, \sum_{j=1}^{m} T_{ij} = a_i\}$. Above optimization often entails substantial computational expenses, and the entropic regularization $H = -\sum_{ij} T_{ij} \ln T_{ij}$ is included to reduce the computational cost while maintaining sufficient smoothness (Cuturi, 2013).

## 4 METHOD

This work follows a reconstruction pipeline, as shown in Fig. 1, which introduces data-adaptive mask modeling during encoding and association prototypes during decoding for tabular AD. Given input samples, we first generate a data-space mask and embed the masked samples with an encoder. Then, we adaptively produce various masks in the projection space according to the discrepancy between features and orthogonal basis vectors. Afterward, the decoder maps multiple masked representations from latent space to data space for reconstruction, among which we learn the normal association prototypes by aligning its distribution to the distribution over shared correlation patterns. Both the discrepancy in projection space and the alignment in decoding stage are formulated as OT problems and integrated with reconstruction loss for optimization and anomaly scoring.

### 4.1 MASKING STRATEGY

Inspired by mask modeling in CV and NLP, we aim to incorporate masks for tabular data to capture intrinsic correlations between features, which facilitates modeling the normal characteristic patterns. However, it is challenging to manually discover such optimal masks. In the following, we introduce the learnable masking strategy both within the raw data space and the projected feature space. The model is motivated to restore the masked features solely relying on the unmasked normal features. Compared to straightforward random masking which may leak a large amount of abnormal information, our data-adaptive masking strategy aims to learn the optimal masks like suspicious anomaly locations. In this way, the reconstructed data are prone to be normal as the suspicious parts are already masked, leading to larger reconstruction errors of anomalies. Thus, we can distinguish the anomaly data by large reconstruction errors deviating from normal ones.

**Data-Space Mask Generalization.** To capture intrinsic correlations existing in the original data space of training data and eliminate redundant information, we produce a learnable soft mask for input data. Given $N$ input samples $\mathbf{X} = \{\boldsymbol{x}_n\}_{n=1}^{N} \in \mathbb{R}^{N \times d}$, the data-adaptive masking can be implemented as

$$\hat{\mathbf{X}} = \mathbf{X} \odot \mathbf{M}^{ds}, \qquad \mathbf{M}^{ds} = \frac{1}{1 + e^{-\mathbf{W}_3(\text{Relu}(\mathbf{W}_2(\text{Relu}(\mathbf{W}_1\mathbf{X}^T))))}}, \tag{3}$$

where $\mathbf{W}_1 \in \mathbb{R}^{d \times d}, \mathbf{W}_2 \in \mathbb{R}^{d \times d}, \mathbf{W}_3 \in \mathbb{R}^{d \times d}$ represent linear projections and $\odot$ refers to the element-wise multiplication. Each value of mask matrix $\mathbf{M}^{ds} \in \mathbb{R}^{N \times d}$ is a flexible weight between zero and one, where each row corresponds to the masking degree across different features, and each column represents the masking degree across input samples for a specific feature. This motivates the model to uncover the statistical correlations between masked and unmasked positions across both datapoints and features. However, the data or feature correlations in tabular data space are often highly tangled and lack statistical global structure information, thus we need to further find another disentangled space for mask modeling.

**Projection-Space Mask Generation.** To stimulate global data correlation learning, we subsequently encode the masked input $\hat{\mathbf{X}}$ into a disentangled Projection Space (P-space) with an NPT layer composed of an ABD and an ABA layer, denoted as $\mathbf{Z} = \boldsymbol{\Phi}_E(\hat{\mathbf{X}}; \boldsymbol{\theta}_E) \in \mathbb{R}^{N \times H}$, where $\boldsymbol{\Phi}_E$ is the encoder parameterized by $\boldsymbol{\theta}_E$, and $\mathbf{Z} = \{\boldsymbol{z}_n\}_{n=1}^N$ where $\boldsymbol{z}_n \in \mathbb{R}^H$ is the representation of $n$-th sample. Intuitively, we only possess the normal tabular samples during training and we assume that they share some global intrinsic characteristic patterns in the P-space. These shared patterns serve as basis vectors and are denoted as $\mathcal{B} = \{\boldsymbol{\beta}^1, ..., \boldsymbol{\beta}^K\}_{k=1}^K \in \mathbb{R}^{K \times H}$, where $K$ is the number of basis vectors and $\boldsymbol{\beta}^k \in \mathbb{R}^H$ denotes the $k^{th}$ basis vector. Typically, normal samples are close to the shared basis vectors, whereas abnormal samples are distinguished by large deviations from these vectors. Thus, we introduce $\mathbf{M} \in \mathbb{R}^{N \times H \times K}$ to mask the suspicious anomaly information in P-space:

$$\mathbf{H}^k = \mathbf{Z} \odot \mathbf{M}^k, \quad M_{nh}^k = \begin{cases} 1, & (z_{nh} - \beta_h^k)^2 \leq \mu_n^k, \\ 0, & (z_{nh} - \beta_h^k)^2 > \mu_n^k, \end{cases} \quad (4)$$

where $\mathbf{H}^k \in \mathbb{R}^{N \times H}$ denotes the masked representation with $k = 1 : K$, $\mu_n^k = \frac{1}{H} \sum_{h=1}^H (z_{nh} - \beta_h^k)^2$ means a data-related threshold computed by the statistic average along the feature dimension, and mask value $M_{nh}^k$ is element-wisely computed by the Euclidean distance between the basis vector $\boldsymbol{\beta}^k$ and latent representation $\boldsymbol{z}_n$ at the $h$-th feature. Intuitively, the positions with larger distances to basis vectors are considered with larger probability as anomalies. By masking these positions, the model is motivated to embed these positions with unmasked normal information, leading to normal reconstructions and large deviations for indicating anomalies. Furthermore, the multiple masking strategy encourages the model to reconstruct samples with various masks. Therefore, anomalies are prone to be detected by a comprehensive measurement. In contrast to masking in the original space, it is more disentangled to act within this P-Space consisting of explicitly defined basis vectors. Furthermore, this designation also saves computational consumption as the multiple setting is only needed for the subsequent decoder.

**Projection Space learning.** In the P-space, we aim to find a group of basis vectors $\mathcal{B}$ to capture the normal characteristics of the training data. We mathematically represent the $K$ basis vectors as a K-dimensional empirical uniform distribution $Q(\mathcal{B}) = \frac{1}{K} \sum_{k=1}^K \delta_{\boldsymbol{\beta}^k}$, where $\delta_{\boldsymbol{\beta}^k}$ is Dirac function of $k^{th}$ basis vectors of the discrete distribution. Besides, we view the P-space representations of $N$ data samples within the training set as another N-dimensional discrete distribution $P(\boldsymbol{\theta}_E) = \frac{1}{N} \sum_{n=1}^N \delta_{\boldsymbol{z}_n}$. Since $\mathcal{B}$ is viewed as the global shared characteristics of the training normal data, we can enforce the distribution $Q(\mathcal{B})$ to approximate the distribution $P(\boldsymbol{\theta}_E)$ to learn the encoder and basis vectors, where we solve the projection space learning problem via distribution matching. Specifically, we first learn the transport plan by minimizing the regularized distance $\mathrm{OT}(P(\boldsymbol{\theta}_E), Q(\mathcal{B}))$ between these two distributions and we design the optimization loss based on the resultant transport plan, stated as

$$\min_{\boldsymbol{\theta}_E, \mathcal{B}} \mathcal{L}_{bv} = \sum_{n=1}^N \min_{k \in K} T_{nk}^{\star} C_{nk}, \quad \text{subject to} \quad \mathbf{T}^{\star} = \argmin_{\mathbf{T} \in \Pi(P(\boldsymbol{\theta}_E), Q(\mathcal{B}))} \langle \mathbf{T}, \mathbf{C} \rangle - \lambda H(\mathbf{T}), \quad (5)$$

where $H(\mathbf{T})$ denotes the regularized entropy in Cuturi (2013), $\lambda > 0$ is the hyper-parameter for the entropy, $\mathbf{C}$ is the transport cost matrix defined as $C_{nk} = \sqrt{(\boldsymbol{z}_n - \boldsymbol{\beta}^k)^2}$, and $\mathbf{T}$ is the transport probability matrix satisfying $\Pi(P(\boldsymbol{\theta}_E), Q(\mathcal{B})) := \{\mathbf{T} \in \mathbb{R}^{N \times K} | \sum_{k=1}^K T_{nk} = \frac{1}{N}, \sum_{n=1}^N T_{nk} = \frac{1}{K}\}$. Notably, during training, we minimize the transport distance from the P-space representation of each sample to its corresponding nearest basis vector based on the learned transport plan. The intuition behind this is normal representations tend to approach specific one of the global basis vectors rather than its fusions, alleviating the potential collapse that anomalous projection representations also exhibit similarity with the fusion version of basis vectors. Since all the training data are normal and

thus the learned basis vectors reflect the normal patterns, at the inference stage, anomalies are prone to deviate from the nominal distributions and thus can be detected by a larger distance even with its corresponding nearest basis vector. Furthermore, the basis vectors in P-space are expected to maintain orthogonal independence from each other, which can be achieved with the soft orthogonality constraint under the standard Frobenius norm, formulated as

$$\min_{\mathcal{B}} \mathcal{L}_{orth} = \min ||\mathbf{B}\mathbf{B}^T - \mathbf{I}||_F^2, \text{ where } \mathbf{B} = \{\boldsymbol{b}_k\}_{k=1}^K \text{ and } \boldsymbol{b}_k = \frac{\boldsymbol{\beta}^k}{||\boldsymbol{\beta}^k||}, k = 1, ..., K. \quad (6)$$

## 4.2 NOMINAL ASSOCIATION PROTOTYPE LEARNING

In this section, we find that learning the typical correlation patterns across features, named association prototypes, facilitates modeling normal characteristic patterns of tabular data. Typically, during training with normal data, we learn the nominal association prototypes from the attention matrix of transformers. During inference, the abnormal associations are different from the nominal association prototypes, which encourages us to incorporate a calibration distance to indicate anomalies. To this end, we learn the association prototypes and the calibration distance by solving a transport distribution matching problem.

**Association Prototypes Learning.** Given the $K$ masked representations $\{\mathbf{H}^k\}_{k=1}^K$ as discussed above, we use the shared decoder to output the corresponding $\{\mathbf{X}_k^{rec}\}_{k=1}^K$ in the data space, denoted as $\mathbf{X}_k^{rec} = \boldsymbol{\Phi}_D(\mathbf{H}^k; \boldsymbol{\theta}_D)$, where the decoder $\boldsymbol{\Phi}_D$ is an NPT layer composed of an ABD and an ABA layer and parameterized by $\boldsymbol{\theta}_D$. To learn the correlation patterns between features in sample $\boldsymbol{x}_n$, we investigate its query $\mathbf{Q} \in \mathbb{R}^{d \times h_k}$ and key $\mathbf{K} \in \mathbb{R}^{d \times h_k}$ matrices for computing attribute attention of $n$-th sample in the ABA layer, where $h_k$ is the latent dimension and $d$ is the feature number of each sample. Note that the query and key matrices can be used to compute the attention matrix in MHSA, denoted as $\mathbf{A} = \text{softmax}(\frac{\mathbf{Q}\mathbf{K}^T}{\sqrt{h_k}}) \in \mathbb{R}^{d \times d}$, which provides a comprehensive understanding of the across-feature association in the $n$-th sample. Here, we establish the lightweight association vector of data $\boldsymbol{x}_n$ to outline the correlations, stated as $\boldsymbol{\pi}_n = \{\pi_n^1, ..., \pi_n^d\} \in \mathbb{R}^d$ with $\pi_n^i = \sum_{j=1}^{h_k} \frac{q_{ij} \cdot k_{ij}}{\sqrt{h_k}}$. Accordingly, we formulate a discrete uniform distribution of all association vectors as $P(\boldsymbol{\pi}) = \sum_{n=1}^N \frac{1}{N} \delta_{\boldsymbol{\pi}_n}$. Besides, we denote $M$ to-be-learned association prototypes as $\Upsilon = \{\boldsymbol{\gamma}^1, ..., \boldsymbol{\gamma}^M\}_{m=1}^M \in \mathbb{R}^{M \times d}$ to extract shared correlation patterns, which form another discrete uniform distribution $Q(\Upsilon) = \sum_{m=1}^M \frac{1}{M} \delta_{\gamma^m}$. Similar to the P-space, we here also enforce the matching between the distribution $P(\boldsymbol{\pi})$ over the association vectors and the distribution $Q(\Upsilon)$ over association prototypes. To this end, we design an OT-based optimization objective by minimizing the transport distance from the association vector to its corresponding nearest association prototype, formulated as

$$\min_{\boldsymbol{\theta}_D, \Upsilon} \mathcal{L}_{ap} = \sum_{n=1}^N \min_{m \in M} \hat{T}_{nm}^\star \hat{C}_{nm}, \quad \text{subject to} \quad \hat{\mathbf{T}}^\star = \underset{\hat{\mathbf{T}} \in \Pi(P(\boldsymbol{\pi}), Q(\Upsilon))}{\arg \min} \langle \hat{\mathbf{T}}, \hat{\mathbf{C}} \rangle - \lambda H(\hat{\mathbf{T}}), \quad (7)$$

where cost matrix $\hat{\mathbf{C}} \in \mathbb{R}^{N \times M}$ is calculated by Euclidean distance and the transport probability matrix $\hat{\mathbf{T}} \in \mathbb{R}^{N \times M}$ satisfy $\Pi(P(\boldsymbol{\pi}), Q(\Upsilon)) := \{\hat{\mathbf{T}} | \sum_{n=1}^N \hat{T}_{nm} = \frac{1}{M}, \sum_{m=1}^M \hat{T}_{nm} = \frac{1}{N}\}$.

**Anomaly Scoring.** The overall algorithm of PTAD is detailed in Appendix A, and the framework follows a reconstruction pipeline which is listed in Appendix B. During inference, the reconstruction loss, typically computed as the point-wise L2 norm, is widely employed as a criterion for anomaly detection. The intuition is that the reconstruction error tends to be higher for anomalous inputs, as the model is solely trained on normal data. In our model, with the $K$ reconstructions recovered corresponding to the multiple P-space masks, we design a more robust and comprehensive reconstruction loss by $\boldsymbol{s}_n^{rec} = \frac{1}{K} \sum_{k=1}^K \|\boldsymbol{x}_n - \boldsymbol{x}_{n,k}^{rec}\|_2^2$. However, relying solely on the reconstruction loss can be suboptimal due to the 'anomaly leakage' issue. This motivates us to propose a new criterion to enhance the discriminability between normal and abnormal samples. In our model, the P-space representation dissimilarity to basis vectors indicates abnormal characteristics, and the association vector dissimilarity to normal association prototypes shows abnormal dependencies. Thus, we refine the anomaly score with the calibration costs $\boldsymbol{s}_n^{ap}$ and $\boldsymbol{s}_n^{bv}$, stated as:

$$\boldsymbol{s}_n^{cab} = \boldsymbol{s}_n^{rec} + \kappa \boldsymbol{s}_n^{bv} + \alpha \boldsymbol{s}_n^{ap}, \quad \boldsymbol{s}_n^{ap} = \min_{m \in M} \hat{T}_{nm}^\star \hat{C}_{nm}, \quad \boldsymbol{s}_n^{bv} = \min_{k \in K} T_{nk}^\star C_{nk}, \quad (8)$$

where $\hat{T}_{nm}^\star$ and $T_{nk}^\star$ subject to Eq. 5 and Eq. 7, respectively, and $\kappa$ and $\alpha$ are weighted coefficients.

## 5 EXPERIMENTS

**Datasets** Following previous work (Yin et al., 2024), we use 20 commonly used tabular datasets spanning multiple domains, including environmental studies, satellite remote sensing, healthcare, finance, etc. Specifically, 12 datasets were sourced from OOD (Rayana, 2016) and 8 from ADBench (Han et al., 2022). Detailed descriptions of these datasets are provided in the appendix C.

**Evaluation metrics** Following the methodology outlined in the literature (Zong et al., 2018; Bergman & Hoshen, 2020), we randomly selected one-half of the normal samples as the training set. The other half of the normal samples are combined with all the anomalous samples to form the test set. We adopted the Area Under the Receiver Operating Characteristic Curve (AUC-ROC) and the Area Under the Precision-Recall Curve (AUC-PR) as our evaluation metrics.

**Implementation details** Both the encoder and decoder contain an NPT architecture, each consisting of an ABD layer and an ABA layer with each attention module containing 4 attention heads. Following Kossen et al. (2021), we utilize a Row-wise feed-forward (rFF) network containing one hidden layer, employing a 4x expansion factor and GeLU activation with the dropout rate of 0.1 for both attention weights and hidden layers. For input and output embeddings, the hidden size of the linear layer to encode the feature is set to 16. For each dataset, we use 5 basis vectors and 5 association prototypes. LAMB (You et al., 2019) with $\beta$ is used as the optimizer including a Lookahead (Zhang et al., 2019) wrapper with update rate $\alpha = 0.5$ and $k = 6$ steps between updates. In the first 10 epochs of training, we apply a warm-up strategy (He et al., 2016; Goyal, 2017) to gradually decrease the learning rate, followed by a cosine annealing strategy (Loshchilov & Hutter, 2016) to adjust the learning rate in subsequent epochs. The whole model is trained end-to-end under the loss $\mathcal{L} = \mathcal{L}_{rec} + \mathcal{L}_{bv} + \mathcal{L}_{ap} + \lambda_{orth}\mathcal{L}_{orth}$, where $\lambda_{orth}$ is set to 0.1. Unless specified otherwise, we set the hyper-parameter of regularized entropy as $\lambda = 0.1$, and score weights $\kappa$ and $\alpha$ are set to 0.01. More details are provided in the appendix D. Meanwhile, the discussions about loss weights are provided in the appendix E.

**Baseline methods** We extensively compare our model with tabular anomaly detection methods including both traditional machine learning and deep learning approaches. The traditional machine learning methods include KNN (Ramaswamy et al., 2000), IForest (Liu et al., 2008), LOF (Breunig et al., 2000), OCSVM (Schölkopf et al., 1999) and GMM (Agarwal, 2007). , and the deep learning methods include LUNAR (Goodge et al., 2022), DeepSVDD (Ruff et al., 2018), GOAD (Bergman & Hoshen, 2020), NeuTralAD (Qiu et al., 2021), ICL (Shenkar & Wolf, 2022), DTE-C (Livernoche et al., 2023), NPT-AD (Thimonier et al., 2024), and MCM (Yin et al., 2024). It is noteworthy that the comparison experiments are based on the comprehensive open-source libraries PYOD (Zhao et al., 2019) (reproduction of KNN IForest, LOF, OCSVM, GMM, LUNAR and DeepSVDD) and DeepOD (Xu et al., 2023; 2024) (reproduction of GOAD, NeuTral, and ICL). The remaining baselines were implemented based on the official open-source code. Furthermore, we also compare the MCM model in combination with the NPT model for comparison. In our experiments, all methods were implemented using consistent dataset splits and preprocessing procedures in line with recent research (Qiu et al., 2021; Shenkar & Wolf, 2022; Yin et al., 2024). We report the average performance over three runs throughout this paper.

### 5.1 MAIN RESULT

The AUC-PR and AUC-ROC results of our method and the other competitors are respectively shown in Fig. 2 and Fig. 4. The average ranking results are also shown in Fig. 3 and Fig. 5, while detailed results on each datasets are listed in Table 21 and Table 22 in the Appendix M, which shows that our method achieves the competitive performances over all datasets. Notably, our method significantly outperforms other methods on several datasets, such as Optdigits and Wbc, leading to 8.12% and 6.56% improvements respectively. Even in cases where our method slightly falls short of the best-performing method, its performance remains commendable, with performance gaps within acceptable ranges. On average, our method achieves around 4% improvement over the second-best comparison method MCM, which demonstrates the effectiveness of our proposed method. The attempt to incorporate the deviation with normal patterns for tabular anomaly detection is feasible. Due to space limitations, we only present AUC-PR results in this table. Furthermore, the comparison results of AUC-ROC are listed in Table 22, in which the overall trends are consistent with those of AUC-PR. As for average results, our approach outperforms all the others and achieves the best

or second-best performance on 14 out of the 20 datasets on AUC-ROC. These evaluation results demonstrate the effectiveness and the generalizability of our model for tabular anomaly detection. In addition, the statistical experimental variance and F1-score are listed in Appendix F. Further experiments on the whole OODs benchmark also demonstrate the effectiveness of our method in the Appendix G. Moreover, the convergence loss are shown in Appendix N.

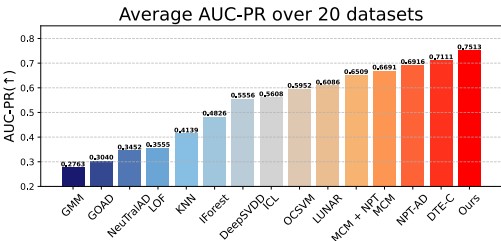

Figure 2: AUC-PR of models over 20 datasets (↑).

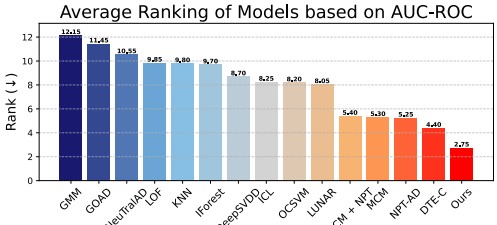

Figure 3: Ranking of model based on AUC-PR (↓).

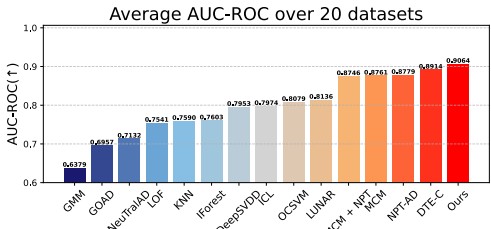

Figure 4: AUC-ROC of models over 20 datasets (↑).

Figure 5: Ranking of model based on AUC-ROC(↓).

## 5.2 ABLATION STUDY

In this section, we explore the effectiveness of different components within PTAD. The average AUC-ROC and AUC-PR across all datasets are reported in Table 1. The variations and observations are listed as follows: **i) Data-space Masking**: Incorporating the data-space mask strategy into the baseline model solely composed of two NPT layers leads to 1.27% improved AUC-PR, demonstrating the effectiveness of soft mask modeling in data space. **ii) Single P-space Learnable Masking**: We generate a single P-space mask and decode masked representation via a single-branch decoder, which results in 3.4% performance gain on AUC-PR, indicating incorporating learnable masks in P space contributes to relieving the anomaly leakage. **iii) Multiple P-space Learnable Masking**: Multiple masks and multiple-branch decoding lead to 4.89% AUC-PR improvement, highlighting the necessity of our multiple designations. **iv) Random Masking**: We randomly generate masks with the same masking rate as our multiple learnable masks. Without the guidance of the basis vectors for generating the mask, the performance degrades by 5.02%, which further confirms the important role of our learnable mask generation method. **v) Association Prototypes**: Incorporating the association prototypes leads to 5.72%, which facilitates us to evaluate the extent of the abnormality by measuring the deviations. **vi) Orthogonality Constrain**: We further validate the orthogonality constrain of basis vectors, the performance gap showcases the orthogonality contributes to anomaly detection by forming disentangled features. **vii)** Overall, the comprehensive version performs best, demonstrating its effectiveness and efficiency as a harmonious combination of its components. The detailed results of the ablation experiments are provided in the Appendix H.

Table 1: Ablation study of our method.

| Data-space Mask | Single Learnable Mask | Multiple Learnable Masks | Random Mask | Association Prototypes | Orthogonality Constrain | AUC-PR | AUC-ROC |
|---|---|---|---|---|---|---|---|
| - | - | - | - | - | - | 0.6381 | 0.8457 |
| ✓ | - | - | - | - | - | 0.6508 | 0.8378 |
| ✓ | ✓ | - | - | - | - | 0.6848 | 0.8835 |
| ✓ | - | ✓ | - | - | ✓ | 0.7337 | 0.8893 |
| ✓ | - | - | ✓ | - | - | 0.6835 | 0.8766 |
| ✓ | - | - | - | ✓ | - | 0.7080 | 0.8884 |
| ✓ | - | ✓ | - | ✓ | - | 0.7392 | 0.8982 |
| ✓ | - | ✓ | - | ✓ | ✓ | **0.7513** | **0.9064** |

## 5.3 ROBUSTNESS TO DIFFERENT TYPES OF ANOMALIES

Although the true distribution of anomalous samples is challenging to capture, previous works (Steinbuss & Böhm, 2021; Han et al., 2022) have identified four common types of anomalies and proposed methods for generating normal and abnormal samples from real datasets. We follow Han et al. (2022) to generate data from the cardiotocography dataset and examine the robustness of our method encountering different anomalous: i) **Local anomalies**: Deviations from their local neighborhoods, generated by scaling the covariance matrix in a Gaussian Mixture Model (GMM). ii) **Global anomalies**: Significantly different from normal data, generated using a uniform distribution based on feature boundaries. iii) **Dependency anomalies**: Samples that violate the dependency structure of normal data, created by enforcing independence among input features. iv) **Clustered anomalies**: Group anomalies with similar characteristics, generated by scaling the mean feature vector of normal samples. More details of anomalies are listed in the Appendix I.

Table 2: Results of four types of anomalies generated from the cardiotocography dataset.

| Category | Metrics | KNN | IForest | LOF | OCSVM | DeepSVDD | GOAD | NeuTralAD | ICL | NPT-AD | MCM | Ours |
|---|---|---|---|---|---|---|---|---|---|---|---|---|
| Local | AUC-PR | 0.2479 | 0.2489 | 0.2611 | 0.2593 | 0.3362 | 0.4797 | 0.4554 | 0.3456 | 0.3326 | 0.5002 | **0.5277** |
| | AUC-ROC | 0.8804 | 0.8657 | 0.8888 | 0.8779 | 0.7849 | 0.8809 | 0.8842 | 0.7138 | 0.4855 | **0.9009** | 0.8907 |
| Global | AUC-PR | 0.3075 | 0.3272 | 0.3009 | 0.3168 | 0.4078 | 0.4663 | 0.4749 | 0.2635 | 0.4731 | 0.5775 | **0.6143** |
| | AUC-ROC | 0.9087 | 0.9157 | 0.9083 | 0.9120 | 0.8823 | 0.9227 | 0.9211 | 0.7104 | 0.9158 | 0.9441 | **0.9464** |
| Dependency | AUC-PR | 0.1895 | 0.1131 | 0.2318 | 0.1160 | 0.2361 | 0.1664 | 0.3350 | 0.2799 | 0.3467 | 0.4495 | **0.4879** |
| | AUC-ROC | 0.8567 | 0.7417 | 0.8918 | 0.7432 | 0.6702 | 0.6773 | 0.8628 | 0.8033 | 0.8364 | 0.9171 | **0.9239** |
| Cluster | AUC-PR | 0.135 | 0.3298 | 0.0718 | 0.2797 | 0.1957 | 0.4748 | 0.1539 | 0.1718 | 0.2021 | 0.5316 | **0.5806** |
| | AUC-ROC | 0.7136 | 0.9136 | 0.4837 | 0.9088 | 0.6806 | 0.9157 | 0.5923 | 0.5266 | 0.6403 | 0.9232 | **0.9391** |

We list the experimental results across the four types of anomalies in Table 2. It can be seen that our model performs well across all four types of anomalies. Especially for dependency and cluster anomalies, our model significantly outperforms the second-best approach, showcasing our model's ability to distinguish these anomalies from normal samples. This might contribute to our special modeling for the dependencies across datapoints and features by multiple masking strategies and the record of both normal features and association patterns.

## 5.4 DISCUSSION

**Computational Cost** Assessing computational cost is crucial for understanding model efficiency and feasibility in practical applications. We present a comprehensive comparison of the computational costs for our method and several strong competitors, including MCM (Yin et al., 2024), NPT-AD (Thimonier et al., 2024), and a combination of MCM and NPT. Four metrics are reported with the average performance across ten times running. As shown in Table 3, we report the computational cost and performances of different models on the campaign dataset. Our method not only ensures strong performance but also maintains a good balance with computational cost.

Table 3: Computational cost on Campaign dataset

| | MCM | NPT-AD | Our method | MCM NPT |
|---|---|---|---|---|
| FLOPS(M) | 3.15 | 3.09 | 9.36 | 46.93 |
| Params(M) | 0.23 | 2.97 | 2.97 | 2.97 |
| Training Time(ms) | 23.35 | 3384.77 | 130.20 | 473.43 |
| Inference Time(ms) | 5.87 | 30904.05 | 22.58 | 201.36 |
| AUC-PR | 0.5543 | 0.4770 | **0.5826** | 0.4954 |
| AUC-ROC | 0.8619 | 0.7915 | **0.8693** | 0.7830 |

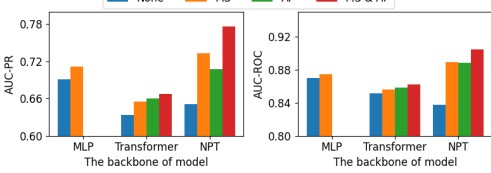

Figure 6: Performance with different backbones

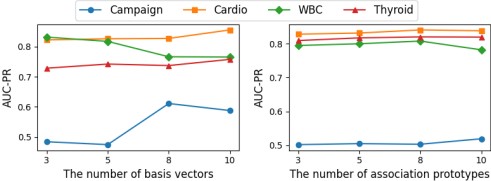

Figure 7: Performance with different numbers

**Different Backbones** The proposed masking strategy and association prototype learning is model-agnostic and flexible, which could be plug-and-played to the other backbone models, such as MLP and vanilla transformers. For MLP, we show the improvement achieved by the masking strategy. For models containing attention maps, such as vanilla transformer and NPT, we demonstrate the performance improvements brought by the two modules respectively and jointly. As illustrated in Fig. 6, the masking strategy (MS) improves model performance across different backbones. For models with attention, i.e. Transformer and NPT, incorporating the association prototype (AP) of

normal samples further enhances the model's ability to distinguish anomalous from normal samples. Furthermore, it is validated that jointly performing both strategies leads to the best performance. Detailed results can be found in the Appendix J.

**Different Number of Basis Vectors & Association Prototypes** Fig. 7 illustrates the impact of varying numbers of basis vectors for masking and association prototypes on four datasets: Campaign, Cardio, WBC, and Thyroid, varying in sample numbers and feature dimensions. It can be seen that as the number of basis vectors increases, the performance across all datasets generally shows an increasing trend. As for the association prototypes, the performance is relatively stable with a slightly increasing trend as the number grows. Therefore, there remains a trade-off between the performance and the number of basis vectors and association prototypes, as a larger number indicates the increased computational cost. Therefore, to achieve a good balance between computational cost and performance, we set the number of both the basis vectors and the association prototypes as 5 in our experiments, aiming to achieve better performances with relatively low costs.

**Distance Measurement** For discussing the distance metric for learning basis vectors and association prototypes, we report the results utilizing MSE distance or OT distance in Table 4. It can be seen that OT distance facilitates optimizing both the basis vector and the association prototype, leading to a large improvement compared to MSE distance. This might contribute to our OT-based method views the points-to-points distance between discrete representations as a transport calibration between two distributions, making a smooth transport and appropriate measurement. Detailed results can be found in the Appendix K.

Table 4: Performances on AUC-PR/AUC-ROC with MSE/OT distances to learn basis vectors (BV) and association prototypes (AP).

|  | AP-MSE | AP-OT |
|---|---|---|
| BV-MSE | 0.6306 / 0.8448 | 0.6543 / 0.8607 |
| BV-OT | 0.6649 / 0.8696 | 0.7513 / 0.9064 |

**Visualization of the P-space Masks** To intuitively understand the P-space masking strategy, we calculate the masking rates of each corresponding feature for visual analysis. As shown in Fig. 8, we selected normal and anomalous sample masks of the Cardio test data with the same average masking rate, i.e. the rates of zeros in both masks are approximately 32%. It can be observed that in the normal sample masks, a high masking rate is only observed for a subset of features, whereas the majority of features exhibit low masking rates, indicating that most of the normal samples are close to the basis vectors of P-space. In contrast, the masking patterns of abnormal samples are quite different, as most features have high masking rates, which illustrate deviations from the normal basis vectors and indicate anomalous. By masking those positions with higher possibilities of abnormality, the decoder is motivated to reconstruct the anomalous samples as normal outputs, leading to larger reconstruction errors for instructing anomalous.

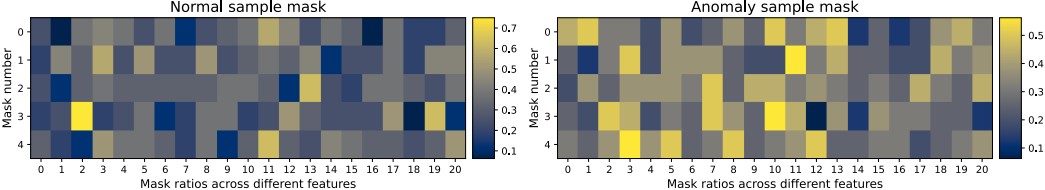

Figure 8: Visualization of P-space masks. The left figure corresponds to the normal sample, and the right figure refers to the abnormal sample, both possessing the same average masking rates.

## 6 CONCLUSION

In this paper, we attribute the 'anomaly leakage' issue in tabular anomaly detection to two main challenges, i.e. representation entanglement and lack of global information. To tackle this problem, we explore mask modeling and prototype learning to enhance anomaly detection performance. The masking modeling involves generating data-adaptive soft masks in data space and multiple learnable masks in disentangled projection space with orthogonal basis vectors. The association prototypes are learned to extract normal characteristic correlations to capture the global data dependencies. Our model is derived from a distribution-matching perspective and formulated as two optimal transport problems, where the calibration costs further refine the anomaly scoring function. The experimental results demonstrated our model's effectiveness, robustness, and generalizability. We hope our way of modeling characteristic patterns of tabular data can potentially extend to wider fields of view.

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

# A   PTAD Algorithm Workflow

The algorithm workflow of our PTAD is listed in Algorithm 1.

---

**Algorithm 1** The Algorithm Workflow of our proposed PTAD.

---

**Input:** Training dataset $D_{train}$;

**Parameters:** NPT Encoder $\boldsymbol{\Phi}_E$, NPT Decoder $\boldsymbol{\Phi}_D$, basis vectors $\mathcal{B} = \{\boldsymbol{\beta}^1, ..., \boldsymbol{\beta}^K\}_{k=1}^K$, association prototypes $\Upsilon = \{\boldsymbol{\gamma}^1, ..., \boldsymbol{\gamma}^M\}_{m=1}^M$, $\{\mathbf{W}_1, \mathbf{W}_2, \mathbf{W}_3\}$ of data-space mask generator;

**Output:** Reconstructions of input tabular data;

1: Initialize model parameters, $\mathcal{B}$ and $\Upsilon$ randomly
2: **for** epoch $1, 2, ..., T$ **do**
3:     Sample batch of $\mathbf{X} \in \mathbb{R}^{N \times d}$ from input datasets $D_{train}$
4:     Generate Data-space masked data $\hat{\mathbf{X}}$ by Eq. 3
5:     Encode the $\hat{\mathbf{X}}$ through $\boldsymbol{\Phi}_E(\hat{\mathbf{X}}; \boldsymbol{\theta}_E)$ as the P-space representation $\mathbf{Z}$
6:     Build distributions for the P-space representations and basis vectors as $P(\boldsymbol{\theta}_E) = \frac{1}{N}\sum_{n=1}^N \delta_{\boldsymbol{z}_n}$ and $Q(\mathcal{B}) = \frac{1}{K}\sum_{k=1}^K \delta_{\boldsymbol{\beta}^k}$
7:     Calculate OT-based distribution matching loss in the P-space as $\mathcal{L}_{bv}$ in Eq. 5 and the orthogonal loss in Eq. 6
8:     Generate Projection-Space mask $\mathbf{M}$ by Eq. 4 and mask the P-space representations by $\mathbf{H}^k = \mathbf{M}^k \odot \mathbf{Z}$
9:     Reconstruct the multiple masked representations in parallel as $\mathbf{X}_k^{rec} = \boldsymbol{\Phi}_D(\mathbf{H}^k; \boldsymbol{\theta}_D)$
10:     Build distributions for association vectors $\boldsymbol{\pi}$ and association prototypes $\Upsilon$ as $P(\boldsymbol{\pi}) = \sum_{n=1}^N \frac{1}{N}\delta_{\boldsymbol{\pi}_n}$ and $Q(\Upsilon) = \sum_{m=1}^M \frac{1}{M}\delta_{\boldsymbol{\gamma}^m}$
11:     Calculate OT-based distribution matching loss of association prototypes as $\mathcal{L}_{ap}$ in Eq. 7 and the multiple reconstruction loss by $\mathcal{L}_{rec} = \frac{1}{K}\sum_{k=1}^K \|\boldsymbol{x}_n - \boldsymbol{x}_{n,k}^{rec}\|_2^2$
12:     Update model parameters by minimizing $\mathcal{L} = \mathcal{L}_{rec} + \mathcal{L}_{bv} + \mathcal{L}_{ap} + \lambda_{orth}\mathcal{L}_{orth}$
13: **end for**

---

# B   Training Pipeline

The training pipeline of PTAD consists of the following steps (Noting that we take the single vector as example, $\boldsymbol{x}, \boldsymbol{m}, \boldsymbol{z}, \boldsymbol{h}$ is referring to each row of the matrix $\mathbf{X}, \mathbf{M}, \mathbf{Z}, \mathbf{H}$ ):

**Data-space Masking:** Following MCM Yin et al. (2024), for numerical features, we use their original scalar values; for categorical features, we use one-hot encoding to represent categorical features. Both the numerical features and one-hot categorical features are concatenated together as $\boldsymbol{x} \in \mathbf{X} \in \mathbb{R}^d$. Our data-space masks are data-adaptively learned for $d$ features as $\boldsymbol{m}^{ds} \in \mathbb{R}^d$ by Eq. 3. We then mask each feature by directly point-wise multiplicating the mask to its corresponding features as $\hat{\boldsymbol{x}} = \boldsymbol{x} \odot \boldsymbol{m}^{ds}$.

**Encoding with an NPT Layer:** The encoded representations of the masked data $\hat{\boldsymbol{x}}$, are processed through learned linear layers to obtain the embedded representations of individual features. These feature embeddings are then passed into an NPT layer, which consists of an ABD layer followed by an ABA layer, resulting in the output $\boldsymbol{z} \in \mathbb{R}^H$. Details can be found in Appendix C3 of NPT Kossen et al. (2021).

**Projection-Space Masking:** The P-space masks are generated by comparing the representation $\boldsymbol{z} \in \mathbb{R}^H$ and each basis vector $\{\boldsymbol{\beta}^k \in \mathbb{R}^H\}_{k=1}^K$ according to Eq. 4, and generate $K$ masks $\{\boldsymbol{m}^k \in \mathbb{R}^H\}_{k=1}^K$. Then we generate $K$ masked P-space representations $\{\boldsymbol{h}^k = \boldsymbol{m}^k \odot \boldsymbol{z} \in \mathbb{R}^H\}_{k=1}^K$. Note we compute the OT loss by solving the OT problem between representation $\boldsymbol{z}$ and $K$ basis vectors by Eq. 5, which can be utilized to optimize the basis vectors.

**Decoding with an NPT Layer:** We parallelly input $K$ masked representations $\{\boldsymbol{h}^k\}_{k=1}^K$ into the decoder (an NPT layer consisting of an ABD and an ABA layer), respectively. The objective of the $K$ branch is the same: reconstructing the original tabular data. The architecture and parameters of the decoder are shared across $K$ branches. During decoding, we obtain the association vector $\boldsymbol{\pi} \in \mathbb{R}^d$ and compute its OT distance with $M$ association prototypes $\{\boldsymbol{\gamma}^m \in \mathbb{R}^d\}_{m=1}^M$ by Eq. 7,

which can be utilized to optimize the association prototypes. To obtain the estimated output features, we also use linear layers to transform the representations back into features, which serves as the inverse process of the input stage.

**Training Parameters:** In this training pipeline, the parameters need to be optimized including: NPT Encoder $\boldsymbol{\Phi}_E$, NPT Decoder $\boldsymbol{\Phi}_D$, basis vectors $\mathcal{B} = \{\boldsymbol{\beta}^1, ..., \boldsymbol{\beta}^K\}_{k=1}^K$, association prototypes $\boldsymbol{\Upsilon} = \{\boldsymbol{\gamma}^1, ..., \boldsymbol{\gamma}^M\}_{m=1}^M$, and $\{\mathbf{W}_1, \mathbf{W}_2, \mathbf{W}_3\}$ of data-space soft mask generator.

## C    DETAILED DATASETS CHARACTERISTICS

Table 5 shows detailed information about the datasets utilized in our experiments, including the total number of samples, data dimensions, and the number of anomalous samples. These datasets include multiple domains, such as environmental studies, satellite remote sensing, healthcare, and so on, mainly sourced from OOD (Rayana, 2016) and ADBench (Han et al., 2022).

Table 5: Details of 20 Datasets.

| Dataset | Samples | Dims | Anomaly |
|---|---|---|---|
| Arrhythmia | 452 | 274 | 66 (14.6%) |
| Breastw | 683 | 9 | 239 (35.0%) |
| Campaign | 41188 | 62 | 4640 (11.3%) |
| Cardio | 1831 | 21 | 176 (9.6%) |
| Cardiotocography | 2114 | 21 | 466 (22.0%) |
| Census | 299285 | 500 | 18568 (6.2%) |
| Fraud | 284807 | 29 | 492 (0.2%) |
| Glass | 214 | 9 | 9 (4.2%) |
| Ionosphere | 351 | 33 | 126 (35.9%) |
| Mammography | 11183 | 6 | 260 (2.3%) |
| NSL-KDD | 148517 | 122 | 77054 (51.8%) |
| Optdigits | 5216 | 64 | 150 (2.9%) |
| Pendigits | 6870 | 16 | 156 (2.3%) |
| Pima | 768 | 8 | 268 (34.9%) |
| Satellite | 6435 | 36 | 2036 (31.7%) |
| Satimage-2 | 5803 | 36 | 71 (1.2%) |
| Shuttle | 49097 | 9 | 3511 (7.1%) |
| Thyroid | 3772 | 6 | 93 (2.5%) |
| Wbc | 278 | 30 | 21 (7.6%) |
| Wine | 129 | 13 | 10 (7.8%) |

## D    MORE DETAILS OF EXPERIMENTAL SETTING

Typically, similar to NPT (Kossen et al., 2021) and NPT-AD (Thimonier et al., 2024), hyperparameter selection was done to obtain the fastest training loss convergence. We set the batch size to 512 during training for almost all datasets, except for the Census dataset's batch size, which is set as 32 due to the memory limitation caused by its large dimension of features. For the learning rate, we select to achieve the fastest loss convergence for each architecture. To constrain the search space for learning rates, we perform a grid search over the range {0.06, 0.04, 0.02, 0.01} and {0.005, 0.001, 0.0005, 0.0001, 0.00001}. The specific hyperparameter settings are summarized in the Table 6. All experiments were conducted on the Ubuntu 20.04.4 LTS operating system, Intel(R) Xeon(R) Gold 5220 CPU @ 2.20GHz with a single NVIDIA A40 48GB GPU and 512GB of RAM. The framework is implemented with Python 3.8.19 and PyTorch 2.0.1. Other key packages include numpy 1.23.5, pandas 2.0.3, and scipy 1.10.1.

Table 6: Datasets hyperparameters. The batch size -1 refers to the input of the entire training set.

| Dataset | Epoch | Batch size | Learning rate |
|---|---|---|---|
| Arrhythmia | 200 | -1 | 0.01 |
| Breastw | 200 | -1 | 0.001 |
| Campaign | 200 | 512 | 0.001 |
| Cardio | 200 | 512 | 0.00001 |
| Cardiotocography | 200 | 512 | 0.001 |
| Census | 50 | 32 | 0.01 |
| Fraud | 200 | 512 | 0.001 |
| Glass | 200 | -1 | 0.0001 |
| Ionosphere | 200 | -1 | 0.01 |
| Mammography | 200 | 512 | 0.02 |
| NSL-KDD | 200 | 512 | 0.01 |
| Optdigits | 200 | 512 | 0.01 |
| Pima | 200 | -1 | 0.01 |
| Pendigits | 200 | 512 | 0.01 |
| Satellite | 200 | 512 | 0.01 |
| Satimage-2 | 200 | 512 | 0.01 |
| Shuttle | 200 | 512 | 0.01 |
| Thyroid | 200 | 512 | 0.01 |
| Wbc | 200 | -1 | 0.00001 |
| Wine | 200 | -1 | 0.00001 |

## E   DIFFERENT WEIGHTS OF ORTHOGONAL LOSS AND ANOMALY SCORES

To investigate the influence of the orthogonal loss and anomaly scores, we illustrate the impact of different weights on AUC-PR and AUC-ROC across four datasets.

For the orthogonal loss, we conduct experiments and illustrate the results in Fig. 9. It can be seen that the performance is stable on WBC and Cardio datasets while sensitive on the other datasets, which might be due to the tradeoff between regularizing the basis vectors to disentangle through orthogonal loss and ensuring an accurate representation of the P-space representation of normal data. To balance two aspects, we choose 0.1 as the weight for the orthogonal loss in our experiments.

Regarding the anomaly score, Fig. 10 displays the results over different weights across four datasets, where we set the same coefficients $\kappa$ and $\alpha$ of calibration distance. It can be seen that the performance is insensitive to the weights of both anomaly scores $s^{bv}$ and $s^{ap}$. To ensure stability in the anomaly detection, we choose 0.01 as our weighting coefficient. By evaluating the average performance of 20 datasets, the calibration distances complement the anomaly score and further enhance model performance.

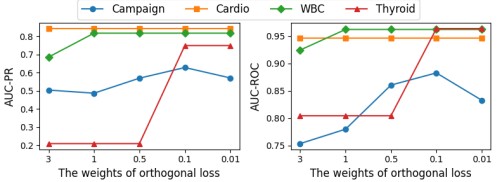

Figure 9: Comparison Results of different weights of orthogonal constrain

Figure 10: Comparison Results of different weights of calibration distances

## F   F1-SCORE AND MORE STATISTICAL EVALUATIONS

In Table 7 and Table 8, we list the F1-score of our model compared with other methods. It can be seen that the F1-score of our model is average better than the comparison methods, which ensures a more robust assessment of our method's performance.

Table 7: F1-score with 5% T-test in 3 runs over 20 datasets

| Dataset | ICL | NPT-AD | PTAD |
|---|---|---|---|
| Arrhythmia | 0.5556±0.0071 | 0.5116±0.0011 | 0.6010±0.0214 |
| Breastw | 0.9066±0.0099 | 0.9581±0.0011 | 0.9749±0.0000 |
| Campaign | 0.4535±0.0226 | 0.5159±0.0033 | 0.5806±0.0439 |
| Cardio | 0.7708±0.0219 | 0.7800±0.0099 | 0.7891±0.0080 |
| Cardiotocography | 0.5551±0.0307 | 0.5902±0.0049 | 0.6652±0.0228 |
| Census | 0.1843±0.0039 | 0.2185±0.0134 | 0.3879±0.0127 |
| Fraud | 0.6375±0.0340 | 0.4924±0.0372 | 0.5659±0.0114 |
| Glass | 0.1481±0.0524 | 0.3112±0.0336 | 0.3333±0.0533 |
| Ionosphere | 0.9339±0.0099 | 0.9459±0.0050 | 0.9133±0.0072 |
| Mammography | 0.1481±0.0765 | 0.4643±0.0045 | 0.4546±0.0054 |
| NSL-KDD | 0.6084±0.0492 | 0.7979±0.0408 | 0.8410±0.1005 |
| Optdigits | 0.5222±0.0974 | 0.5200±0.0757 | 0.7133±0.0565 |
| Pima | 0.6057±0.0307 | 0.7369±0.0052 | 0.6778±0.0162 |
| Pendigits | 0.4979±0.1282 | 0.8907±0.0217 | 0.8462±0.0388 |
| Satellite | 0.7698±0.0121 | 0.7640±0.0060 | 0.7205±0.0080 |
| Satimage-2 | 0.7934±0.0266 | 0.9614±0.0039 | 0.9343±0.0150 |
| Shuttle | 0.9804±0.0009 | 0.9836±0.0013 | 0.9658±0.0030 |
| Thyroid | 0.6953±0.0623 | 0.7249±0.0038 | 0.7092±0.0150 |
| Wbc | 0.4667±0.0943 | 0.7523±0.0036 | 0.7424±0.0539 |
| Wine | 0.1333±0.1247 | 0.8172±0.0154 | 0.9091±0.0000 |
| Average | 0.5683±0.0447 | 0.6868±0.0145 | 0.7162±0.0246 |

Table 8: F1-score with 5% T-test in 20 runs over 16 datasets

| Dataset | ICL | NPT-AD | PTAD |
|---|---|---|---|
| Arrhythmia | 0.5659±0.0226 | 0.5069±0.0018 | 0.5811±0.0119 |
| Breastw | 0.9042±0.0234 | 0.9584±0.0006 | 0.9749±0.0000 |
| Campaign | 0.4485±0.0159 | 0.5178±0.0025 | 0.5781±0.0218 |
| Cardio | 0.7560±0.0342 | 0.7849±0.0026 | 0.7855±0.0112 |
| Cardiotocography | 0.5476±0.0353 | 0.5854±0.0033 | 0.6601±0.0160 |
| Glass | 0.1278±0.0530 | 0.3081±0.0134 | 0.2750±0.0235 |
| Ionosphere | 0.9139±0.0187 | 0.9470±0.0016 | 0.9078±0.0019 |
| Mammography | 0.2067±0.0774 | 0.4634±0.0029 | 0.4507±0.0078 |
| Optdigits | 0.4823±0.1261 | 0.2546±0.0055 | 0.6233±0.0274 |
| Pima | 0.6093±0.0223 | 0.7371±0.0015 | 0.6688±0.0100 |
| Pendigits | 0.4971±0.1320 | 0.8983±0.0062 | 0.8522±0.0127 |
| Satellite | 0.7659±0.0164 | 0.7668±0.0031 | 0.7129±0.0047 |
| Satimage-2 | 0.8007±0.0725 | 0.9609±0.0020 | 0.9331±0.0047 |
| Thyroid | 0.7054±0.0516 | 0.7271±0.0022 | 0.7026±0.0068 |
| Wbc | 0.4900±0.1091 | 0.7476±0.0044 | 0.7197±0.0228 |
| Wine | 0.3250±0.1894 | 0.8222±0.0153 | 0.8954±0.0190 |
| Average | 0.5716±0.0624 | 0.6866±0.0043 | 0.7076±0.0126 |

In Table 9, Table 10, Table 11, and Table 12, we list the statistical evaluations of AUC-PR and AUR-ROC over 3 and 20 runs. It can be seen that the variance of multiple runs of our model is comparable with the comparison methods, which showcases that our model is robust and stable across multiple runs.

Table 9: AUC-PR with 5% T-test in 3 runs over 20 datasets

| Dataset | ICL | NPT-AD | PTAD |
|---|---|---|---|
| Arrhythmia | 0.5773±0.0008 | 0.4779±0.0078 | 0.6164±0.0074 |
| Breastw | 0.9459±0.0071 | 0.9815±0.0016 | 0.9973±0.0001 |
| Campaign | 0.4291±0.0241 | 0.4852±0.0118 | 0.5826±0.0313 |
| Cardio | 0.8054±0.0185 | 0.8216±0.0057 | 0.8445±0.0063 |
| Cardiotocography | 0.6443±0.0183 | 0.6443±0.0046 | 0.6962±0.0429 |
| Census | 0.1850±0.0046 | 0.2363±0.0274 | 0.2970±0.0162 |
| Fraud | 0.5909±0.0213 | 0.3972±0.0440 | 0.5377±0.0087 |
| Glass | 0.2296±0.0239 | 0.2235±0.0215 | 0.3880±0.0456 |
| Ionosphere | 0.9771±0.0055 | 0.9875±0.0031 | 0.9813±0.0006 |
| Mammography | 0.1792±0.0394 | 0.4133±0.0024 | 0.4398±0.0036 |
| NSL-KDD | 0.5621±0.0280 | 0.8603±0.0154 | 0.8823±0.0413 |
| Optdigits | 0.4400±0.0977 | 0.1251±0.0026 | 0.7957±0.0852 |
| Pima | 0.6462±0.0187 | 0.6858±0.0037 | 0.7308±0.0071 |
| Pendigits | 0.4003±0.1289 | 0.9388±0.0255 | 0.9260±0.0206 |
| Satellite | 0.8976±0.0098 | 0.8540±0.0075 | 0.8433±0.0016 |
| Satimage-2 | 0.8599±0.0547 | 0.9859±0.0005 | 0.9844±0.0003 |
| Shuttle | 0.9766±0.0029 | 0.9656±0.0005 | 0.9377±0.0086 |
| Thyroid | 0.6834±0.0713 | 0.7851±0.0029 | 0.7685±0.0033 |
| Wbc | 0.7795±0.1255 | 0.7497±0.0045 | 0.8451±0.0198 |
| Wine | 0.3631±0.0260 | 0.7635±0.0108 | 0.9323±0.0012 |
| Average | 0.6086±0.0363 | 0.6691±0.0144 | 0.7513±0.0175 |

Table 10: AUC-ROC with 5% T-test in 3 runs over 20 datasets

| Dataset | ICL | NPT-AD | PTAD |
|---|---|---|---|
| Arrhythmia | 0.7937±0.0022 | 0.7103±0.0055 | 0.8147±0.0089 |
| Breastw | 0.9622±0.0064 | 0.9834±0.0010 | 0.9973±0.0001 |
| Campaign | 0.7032±0.0316 | 0.7778±0.0008 | 0.8693±0.0281 |
| Cardio | 0.9140±0.0101 | 0.9211±0.0032 | 0.9653±0.0019 |
| Cardiotocography | 0.6840±0.0140 | 0.6840±0.0071 | 0.8210±0.0305 |
| Census | 0.6725±0.0097 | 0.7008±0.0469 | 0.7622±0.0097 |
| Fraud | 0.9143±0.0016 | 0.9564±0.0031 | 0.9531±0.0095 |
| Glass | 0.8196±0.0280 | 0.7843±0.0316 | 0.8353±0.0305 |
| Ionosphere | 0.9741±0.0045 | 0.9805±0.0029 | 0.9738±0.0008 |
| Mammography | 0.5653±0.0577 | 0.8928±0.0013 | 0.8882±0.0024 |
| NSL-KDD | 0.2665±0.0905 | 0.8126±0.0468 | 0.8513±0.0982 |
| Optdigits | 0.9552±0.0257 | 0.8084±0.0307 | 0.9825±0.0038 |
| Pima | 0.6231±0.0247 | 0.7161±0.0044 | 0.7234±0.0177 |
| Pendigits | 0.8334±0.0771 | 0.9983±0.0009 | 0.9961±0.0014 |
| Satellite | 0.8805±0.0123 | 0.7914±0.0154 | 0.7992±0.0038 |
| Satimage-2 | 0.9828±0.0121 | 0.9995±0.0000 | 0.9995±0.0001 |
| Shuttle | 0.9889±0.0019 | 0.9986±0.0000 | 0.9965±0.0002 |
| Thyroid | 0.9223±0.0225 | 0.9762±0.0010 | 0.9750±0.0039 |
| Wbc | 0.9087±0.0214 | 0.9577±0.0033 | 0.9737±0.0126 |
| Wine | 0.7927±0.0205 | 0.9567±0.0032 | 0.9507±0.0038 |
| Average | 0.8078±0.0237 | 0.8779±0.0104 | 0.9064±0.0133 |

Table 11: AUC-PR with 5% T-test in 20 runs over 16 datasets

| Dataset | ICL | NPT-AD | PTAD |
|---|---|---|---|
| Arrhythmia | 0.5877±0.0163 | 0.4270±0.0027 | 0.5929±0.0060 |
| Breastw | 0.9429±0.0235 | 0.9841±0.0006 | 0.9972±0.0001 |
| Campaign | 0.4281±0.0172 | 0.4861±0.0035 | 0.5595±0.0255 |
| Cardio | 0.8172±0.0273 | 0.7928±0.0018 | 0.8425±0.0099 |
| Cardiotocography | 0.6554±0.0320 | 0.6430±0.0034 | 0.6927±0.0163 |
| Glass | 0.2277±0.0293 | 0.2688±0.0065 | 0.3147±0.0274 |
| Ionosphere | 0.9750±0.0061 | 0.9759±0.0011 | 0.9783±0.0009 |
| Mammography | 0.1902±0.0542 | 0.4043±0.0021 | 0.4642±0.0062 |
| Optdigits | 0.4237±0.1205 | 0.2103±0.0713 | 0.6305±0.0334 |
| Pima | 0.6559±0.0239 | 0.6885±0.0012 | 0.7125±0.0106 |
| Pendigits | 0.4534±0.1342 | 0.9451±0.0072 | 0.9198±0.0091 |
| Satellite | 0.8925±0.0134 | 0.8563±0.0026 | 0.8401±0.0039 |
| Satimage-2 | 0.8371±0.0949 | 0.9862±0.0003 | 0.9730±0.0026 |
| Thyroid | 0.6729±0.0635 | 0.7902±0.012 | 0.7460±0.0048 |
| Wbc | 0.5171±0.1103 | 0.7826±0.0031 | 0.8234±0.0145 |
| Wine | 0.3776±0.1212 | 0.8812±0.0111 | 0.9257±0.0068 |
| Average | 0.6034±0.0554 | 0.6951±0.0081 | 0.7508±0.0111 |

Table 12: AUC-ROC with 5% T-test in 20 runs over 16 datasets

| Dataset | ICL | NPT-AD | PTAD |
|---|---|---|---|
| Arrhythmia | 0.8040±0.0133 | 0.7110±0.0014 | 0.7915±0.0067 |
| Breastw | 0.9566±0.0156 | 0.9853±0.0004 | 0.9971±0.0001 |
| Campaign | 0.7032±0.0173 | 0.7935±0.0028 | 0.8428±0.0213 |
| Cardio | 0.9178±0.0156 | 0.9488±0.0011 | 0.9637±0.0030 |
| Cardiotocography | 0.6934±0.0382 | 0.7184±0.0043 | 0.8140±0.0122 |
| Glass | 0.8266±0.0197 | 0.7681±0.0134 | 0.7559±0.0248 |
| Ionosphere | 0.9719±0.0072 | 0.9687±0.0011 | 0.9696±0.0015 |
| Mammography | 0.5705±0.0824 | 0.8891±0.0012 | 0.8901±0.0024 |
| Optdigits | 0.9412±0.0295 | 0.7729±0.0631 | 0.9758±0.0026 |
| Pima | 0.6248±0.0217 | 0.7215±0.0011 | 0.7013±0.0115 |
| Pendigits | 0.8887±0.0691 | 0.9976±0.0003 | 0.9945±0.0010 |
| Satellite | 0.8754±0.0161 | 0.8042±0.0051 | 0.7897±0.0086 |
| Satimage-2 | 0.9805±0.0107 | 0.9995±0.0000 | 0.9973±0.0009 |
| Thyroid | 0.9245±0.0245 | 0.9704±0.0003 | 0.9691±0.0016 |
| Wbc | 0.9323±0.0247 | 0.9531±0.0014 | 0.9557±0.0052 |
| Wine | 0.8159±0.0661 | 0.9758±0.0025 | 0.9455±0.0053 |
| Average | 0.8392±0.0294 | 0.8736±0.0062 | 0.8971±0.0067 |

## G  COMPARE ON OODS BENCHMARK

We have conducted additional experiments on the full set of 30 ODDS datasets to provide a more comprehensive evaluation of our method. Detailed results are listed in Table 13 and Table 14 by reporting the AUC-PR and AUC-ROC, which consistently showcase our superiority compared to other methods.

Table 13: Compare with other baselines in AUC-PR on OODS

| Dataset | GMM | LUNAR | NPT-AD | DDPM | DTE-IG | DTE-C | PTAD |
|---|---|---|---|---|---|---|---|
| Annthyroid | 0.1414 | 0.1685 | 0.6457 | 0.4689 | 0.3873 | 0.8314 | 0.5464 |
| Arrhythmia | 0.3071 | 0.3602 | 0.4779 | 0.5660 | 0.6252 | 0.6609 | 0.6164 |
| Breastw | 0.9490 | 0.9047 | 0.9815 | 0.9743 | 0.7894 | 0.9207 | 0.9973 |
| Cardio | 0.3373 | 0.1733 | 0.8216 | 0.5667 | 0.6239 | 0.7125 | 0.8445 |
| Ecoli | 0.6532 | 0.3749 | 0.8013 | 0.3932 | 0.3668 | 0.7136 | 0.7187 |
| Forest | 0.0802 | 0.0355 | 0.0184 | 0.0527 | 0.8887 | 0.7148 | 0.0309 |
| Glass | 0.0586 | 0.1146 | 0.2235 | 0.3102 | 0.7224 | 0.5002 | 0.3880 |
| Http | 0.2604 | 0.0031 | 0.9371 | 0.9941 | 0.9837 | 0.8967 | 0.7095 |
| Ionosphere | 0.9589 | 0.9511 | 0.9875 | 0.9557 | 0.9821 | 0.9801 | 0.9813 |
| Letter | 0.2510 | 0.2805 | 0.7683 | 0.0934 | 0.1576 | 0.0960 | 0.2016 |
| Lympho | 0.6625 | 0.7917 | 0.9920 | 0.6272 | 0.8129 | 0.7314 | 0.7843 |
| Mammography | 0.1941 | 0.1396 | 0.4133 | 0.1268 | 0.1450 | 0.4045 | 0.4398 |
| Mnist | 0.3716 | 0.2818 | 0.7648 | 0.5210 | 0.6345 | 0.5650 | 0.7576 |
| Mulcross | 0.9259 | 0.0516 | 1.0000 | 0.9967 | 1.0000 | 1.0000 | 1.0000 |
| Musk | 0.8940 | 0.1432 | 1.0000 | 0.9964 | 1.0000 | 1.0000 | 0.9989 |
| Optdigits | 0.0317 | 0.0321 | 0.1251 | 0.0807 | 0.3386 | 0.1465 | 0.7947 |
| Pendigits | 0.0508 | 0.0557 | 0.9388 | 0.2736 | 0.5389 | 0.4712 | 0.9260 |
| Pima | 0.4873 | 0.5044 | 0.6858 | 0.5980 | 0.6243 | 0.6075 | 0.7308 |
| Satellite | 0.5092 | 0.4513 | 0.8540 | 0.8302 | 0.8857 | 0.8496 | 0.8433 |
| Satimage-2 | 0.4028 | 0.2742 | 0.9859 | 0.7654 | 0.7869 | 0.7473 | 0.9844 |
| Seismic | 0.0865 | 0.0907 | 0.2042 | 0.1480 | 0.1141 | 0.1247 | 0.2419 |
| Shuttle | 0.8623 | 0.1816 | 0.9656 | 0.9754 | 0.9981 | 0.9430 | 0.9337 |
| Smtp | 0.1728 | 0.0242 | 0.5135 | 0.3538 | 0.0089 | 0.5171 | 0.6203 |
| Speech | 0.0217 | 0.0237 | 0.0610 | 0.0384 | 0.0273 | 0.0293 | 0.0352 |
| Thyroid | 0.1779 | 0.1474 | 0.7851 | 0.7653 | 0.3223 | 0.8338 | 0.7685 |
| Vertebral | 0.1012 | 0.0949 | 0.2278 | 0.2432 | 0.2688 | 0.3125 | 0.7458 |
| Vowels | 0.2538 | 0.6250 | 0.9193 | 0.1861 | 0.4039 | 0.3497 | 0.4663 |
| Wbc | 0.4581 | 0.5623 | 0.7497 | 0.6543 | 0.8469 | 0.6453 | 0.8451 |
| Wine | 0.1095 | 0.0585 | 0.7635 | 0.4677 | 1.0000 | 0.8573 | 0.9323 |
| Yeast | 0.3139 | 0.3184 | 0.2239 | 0.4917 | 0.5184 | 0.5021 | 0.5353 |
| Average | 0.3695 | 0.2740 | 0.6612 | 0.5172 | 0.5934 | 0.6221 | **0.6806** |

Table 14: Compare with other baselines in AUC-ROC on OODS

| Datasets | GMM | LUNAR | NPT-AD | DDPM | DTE-IG | DTE-C | PTAD |
|---|---|---|---|---|---|---|---|
| Annthyroid | 0.6292 | 0.7346 | 0.8783 | 0.7474 | 0.7081 | 0.9761 | 0.8596 |
| Arrhythmia | 0.7564 | 0.8297 | 0.7103 | 0.7331 | 0.7782 | 0.8816 | 0.8147 |
| Breastw | 0.9690 | 0.9711 | 0.9834 | 0.9687 | 0.7530 | 0.9360 | 0.9973 |
| Cardio | 0.8703 | 0.5236 | 0.9211 | 0.7437 | 0.8751 | 0.8889 | 0.9653 |
| Ecoli | 0.9177 | 0.7774 | 0.8647 | 0.9012 | 0.6957 | 0.8897 | 0.8791 |
| Forest | 0.9327 | 0.7467 | 0.5392 | 0.7298 | 0.9859 | 0.9742 | 0.6625 |
| Glass | 0.5064 | 0.8462 | 0.7843 | 0.7434 | 0.9564 | 0.9390 | 0.8353 |
| Http | 0.9961 | 0.1823 | 0.9997 | 0.9969 | 0.9999 | 0.9993 | 0.9986 |
| Ionosphere | 0.9741 | 0.9642 | 0.9805 | 0.9357 | 0.9713 | 0.9713 | 0.9738 |
| Letter | 0.8304 | 0.8778 | 0.9597 | 0.3913 | 0.5079 | 0.3749 | 0.6568 |
| Lympho | 0.9792 | 0.9931 | 0.9992 | 0.8570 | 0.9510 | 0.9637 | 0.9765 |
| Mammography | 0.8671 | 0.8323 | 0.8928 | 0.7367 | 0.7937 | 0.8680 | 0.8361 |
| Mnist | 0.8491 | 0.7357 | 0.9464 | 0.7948 | 0.8518 | 0.8731 | 0.9209 |
| Mulcross | 0.9977 | 0.0012 | 1.0000 | 0.9992 | 1.0000 | 1.0000 | 1.0000 |
| Musk | 0.9954 | 0.6020 | 1.0000 | 0.9997 | 1.0000 | 1.0000 | 0.9999 |
| Optdigits | 0.5478 | 0.4530 | 0.8084 | 0.6552 | 0.9193 | 0.8254 | 0.9825 |
| Pendigits | 0.7546 | 0.6835 | 0.9983 | 0.8522 | 0.9759 | 0.9769 | 0.9961 |
| Pima | 0.6529 | 0.6755 | 0.7161 | 0.5506 | 0.6167 | 0.6124 | 0.7234 |
| Satellite | 0.6394 | 0.6213 | 0.7914 | 0.7777 | 0.8618 | 0.7932 | 0.7992 |
| Satimage-2 | 0.9853 | 0.8245 | 0.9995 | 0.9876 | 0.9796 | 0.9951 | 0.9995 |
| Seismic | 0.6032 | 0.6191 | 0.6943 | 0.5100 | 0.4862 | 0.4701 | 0.6558 |
| Shuttle | 0.9809 | 0.6331 | 0.9986 | 0.9980 | 0.9999 | 0.9976 | 0.9965 |
| Smtp | 0.7280 | 0.8857 | 0.8351 | 0.9292 | 0.8003 | 0.9588 | 0.8486 |
| Speech | 0.5452 | 0.5755 | 0.5872 | 0.4748 | 0.4048 | 0.3898 | 0.4398 |
| Thyroid | 0.9207 | 0.8825 | 0.9762 | 0.9556 | 0.9215 | 0.9896 | 0.9750 |
| Vertebral | 0.4368 | 0.3822 | 0.5038 | 0.5012 | 0.5298 | 0.6426 | 0.9079 |
| Vowels | 0.9038 | 0.9502 | 0.9938 | 0.6879 | 0.8771 | 0.8627 | 0.8654 |
| Wbc | 0.9448 | 0.9418 | 0.9577 | 0.9048 | 0.9832 | 0.9681 | 0.9737 |
| Wine | 0.6867 | 0.3800 | 0.9567 | 0.7770 | 1.0000 | 0.9864 | 0.9507 |
| Yeast | 0.4461 | 0.4382 | 0.5038 | 0.4770 | 0.5016 | 0.4715 | 0.5294 |
| Average | 0.7949 | 0.6855 | 0.8594 | 0.7772 | 0.8229 | 0.8492 | **0.8673** |

# H    Detailed Results of Ablation Study

Table 15 and Table 16 show the detailed AUC-PR and AUC-ROC results of the ablation study of our method, including different variations of our model: The two-NPT-layer Baseline, i) Data-space Masking, ii) Single P-space Learnable Masking, iii) Multiple P-space Learnable Masking, iv) Random Masking, v) Association Prototypes, vi) Orthogonality Constrain, and vii) Overall performance. Each component is critical for enhancing anomaly detection, and the comprehensive version performs best, demonstrating its effectiveness as a harmonious combination of its components.

Table 15: Detailed Comparison Results of AUC-PR for Ablation Study

| Dataset | Baseline | i | ii | iii | iv | v | vi | vii |
|---|---|---|---|---|---|---|---|---|
| Arrhythmia | 0.5999 | 0.6113 | 0.6063 | 0.6183 | 0.6005 | 0.6240 | 0.6149 | 0.6164 |
| Breastw | 0.9974 | 0.9977 | 0.9977 | 0.9977 | 0.9977 | 0.9975 | 0.9973 | 0.9973 |
| Campaign | 0.3803 | 0.4458 | 0.4609 | 0.6280 | 0.5429 | 0.5048 | 0.5297 | 0.5826 |
| Cardio | 0.5272 | 0.8216 | 0.8326 | 0.8424 | 0.7350 | 0.8318 | 0.8450 | 0.8445 |
| Cardiotocography | 0.5857 | 0.7103 | 0.7035 | 0.6923 | 0.6752 | 0.7318 | 0.7112 | 0.6962 |
| Census | 0.1990 | 0.2592 | 0.1737 | 0.2315 | 0.2145 | 0.3193 | 0.2918 | 0.2970 |
| Fraud | 0.6088 | 0.4483 | 0.5297 | 0.5538 | 0.4814 | 0.5012 | 0.4869 | 0.5377 |
| Glass | 0.1249 | 0.1496 | 0.3343 | 0.3084 | 0.1263 | 0.2141 | 0.3816 | 0.3880 |
| Ionosphere | 0.9740 | 0.9822 | 0.9759 | 0.9818 | 0.9790 | 0.9746 | 0.9806 | 0.9813 |
| Mammography | 0.2587 | 0.1700 | 0.2714 | 0.4175 | 0.3455 | 0.3821 | 0.4401 | 0.4398 |
| NSL-KDD | 0.8738 | 0.8517 | 0.8837 | 0.8767 | 0.9142 | 0.8572 | 0.8813 | 0.8823 |
| Optdigits | 0.1283 | 0.1476 | 0.5787 | 0.6654 | 0.4383 | 0.5548 | 0.7477 | 0.7957 |
| Pima | 0.7047 | 0.6997 | 0.6980 | 0.7005 | 0.6720 | 0.7620 | 0.7287 | 0.7308 |
| Pendigits | 0.5096 | 0.4889 | 0.6296 | 0.9259 | 0.8242 | 0.7394 | 0.9295 | 0.9260 |
| Satellite | 0.8410 | 0.8349 | 0.8794 | 0.8351 | 0.8317 | 0.8490 | 0.8222 | 0.8433 |
| Satimage-2 | 0.9841 | 0.9830 | 0.9782 | 0.9817 | 0.9880 | 0.8593 | 0.9831 | 0.9844 |
| Shuttle | 0.9119 | 0.9035 | 0.9143 | 0.9195 | 0.9175 | 0.9085 | 0.9324 | 0.9377 |
| Thyroid | 0.8026 | 0.7922 | 0.5036 | 0.7490 | 0.6922 | 0.8175 | 0.7616 | 0.7685 |
| Wbc | 0.8233 | 0.7879 | 0.8013 | 0.8172 | 0.7628 | 0.8001 | 0.8397 | 0.8451 |
| Wine | 0.9276 | 0.9308 | 0.9424 | 0.9308 | 0.9313 | 0.9308 | 0.8784 | 0.9323 |
| Average | 0.6381 | 0.6508 | 0.6848 | 0.7337 | 0.6835 | 0.7080 | 0.7392 | **0.7513** |

Table 16: Detailed Comparison Results of AUC-ROC for Ablation Study

| Dataset | baseline | i | ii | iii | iv | v | vi | vii |
|---|---|---|---|---|---|---|---|---|
| Arrhythmia | 0.8073 | 0.8130 | 0.7998 | 0.8195 | 0.8062 | 0.8281 | 0.8145 | 0.8147 |
| Breastw | 0.9975 | 0.9977 | 0.9977 | 0.9977 | 0.9977 | 0.9975 | 0.9973 | 0.9973 |
| Campaign | 0.7005 | 0.7245 | 0.7791 | 0.8827 | 0.8503 | 0.7936 | 0.8270 | 0.8693 |
| Cardio | 0.8095 | 0.9552 | 0.9603 | 0.9465 | 0.8893 | 0.9609 | 0.9627 | 0.9653 |
| Cardiotocography | 0.6986 | 0.7936 | 0.7861 | 0.8122 | 0.8094 | 0.8420 | 0.8298 | 0.8210 |
| Census | 0.7185 | 0.7112 | 0.6849 | 0.6494 | 0.7200 | 0.7864 | 0.7287 | 0.7622 |
| Fraud | 0.8438 | 0.9239 | 0.9276 | 0.9316 | 0.9011 | 0.9389 | 0.9271 | 0.9531 |
| Glass | 0.6294 | 0.4392 | 0.7637 | 0.7500 | 0.6147 | 0.6461 | 0.7755 | 0.8353 |
| Ionosphere | 0.9624 | 0.9759 | 0.9677 | 0.9750 | 0.9703 | 0.9620 | 0.9729 | 0.9738 |
| Mammography | 0.8232 | 0.6813 | 0.8088 | 0.8764 | 0.8669 | 0.8354 | 0.8885 | 0.8882 |
| NSL-KDD | 0.8695 | 0.8239 | 0.8267 | 0.8525 | 0.8883 | 0.8437 | 0.8698 | 0.8513 |
| Optdigits | 0.7569 | 0.8222 | 0.9757 | 0.9805 | 0.9444 | 0.9699 | 0.9800 | 0.9825 |
| Pima | 0.7161 | 0.6627 | 0.7122 | 0.6598 | 0.6566 | 0.7352 | 0.7726 | 0.7234 |
| Pendigits | 0.9400 | 0.7822 | 0.9835 | 0.9962 | 0.9921 | 0.9716 | 0.9963 | 0.9961 |
| Satellite | 0.7892 | 0.7940 | 0.8558 | 0.7889 | 0.7705 | 0.7947 | 0.7636 | 0.7992 |
| Satimage-2 | 0.9994 | 0.9992 | 0.9987 | 0.9989 | 0.9955 | 0.9961 | 0.9994 | 0.9995 |
| Shuttle | 0.9955 | 0.9951 | 0.9952 | 0.9969 | 0.9971 | 0.9940 | 0.9963 | 0.9965 |
| Thyroid | 0.9651 | 0.9677 | 0.9117 | 0.9636 | 0.9615 | 0.9770 | 0.9619 | 0.9750 |
| Wbc | 0.9579 | 0.9479 | 0.9632 | 0.9620 | 0.9515 | 0.9492 | 0.9530 | 0.9737 |
| Wine | 0.9337 | 0.9461 | 0.9707 | 0.9461 | 0.9476 | 0.9461 | 0.9461 | 0.9507 |
| Average | 0.8457 | 0.8378 | 0.8835 | 0.8893 | 0.8766 | 0.8884 | 0.8982 | **0.9064** |

## I  MORE DETAILS OF PROCESSING FOUR TYPES ANOMALIES

Here, we provide additional details on the generation process of the four types of anomalies, following Han et al. (2022).

- **Local anomalies:** The classic GMM procedure (Milligan, 1985; Steinbuss & Böhm, 2021) is used to generate normal samples, after which a covariance scaling parameter $\alpha = 5$ is used to generate anomalous samples.

- **Global anomalies:** The global anomalies are generated from a uniform distribution $Unif(\alpha \cdot \min(\mathbf{X}^k), \alpha \cdot \max(\mathbf{X}^k))$, where the boundaries are defined as the $\min$ and $\max$ of an input feature, such as k-th feature $\mathbf{X}^k$. The hyperparameter $\alpha$ is established at 1.1, influencing the level of deviation exhibited by the anomalies.

- **Dependency anomalies:** For generating independent types of anomalies, Vine Copula method (Aas et al., 2009) is utilized to model the dependency structure of the original data, whereby the probability density function of the generated anomalies is established as completely independent by eliminating the modeled dependencies, which could refer to (Martinez-Guerra & Mata-Machuca, 2014). We use Kernel Density Estimation(KDE) (Hastie et al., 2009) to estimate the probability density function of features and generate normal samples.

- **Clustered anomalies:** We scale the mean feature vector of normal samples by $\alpha = 5$, such as $\hat{\mu} = \alpha\hat{\mu}$. The hyperparameter $\alpha$ scales GMM, controlling the distance between anomaly clusters and the normal for generating anomalies.

## J  DETAILED RESULTS OF DIFFERENT BACKBONES

We present the performances by adding our multiple strategy and association prototype to different backbone models, specifically including a comparison of the performance of MLP with/without the mask strategy, Transformer with/without the mask strategy or association prototype. Table 17 and Table 18 show the AUC-PR and AUC-ROC. The results show that our proposed masking strategy and association prototype learning is model-agnostic and flexible, and can act as a plug-and-play framework and possess good generalizability to other models.

Table 17: Detailed Comparison Results of AUC-PR with different backbones

| Dataset | MLP | MLP MS | Transformer | Transformer MS | Transformer AP | Transformer MS&AP |
|---|---|---|---|---|---|---|
| Arrhythmia | 0.5821 | 0.5689 | 0.5454 | 0.5614 | 0.5878 | 0.5850 |
| Breastw | 0.9977 | 0.9977 | 0.9969 | 0.9970 | 0.9980 | 0.9974 |
| Campaign | 0.5627 | 0.5096 | 0.4869 | 0.5622 | 0.4552 | 0.5163 |
| Cardio | 0.8516 | 0.8521 | 0.8043 | 0.8389 | 0.7134 | 0.8506 |
| Cardiotocography | 0.6919 | 0.7044 | 0.6304 | 0.5928 | 0.7257 | 0.7289 |
| Census | 0.2050 | 0.2120 | 0.1626 | 0.2191 | 0.1595 | 0.1146 |
| Fraud | 0.7404 | 0.6512 | 0.2942 | 0.3535 | 0.3850 | 0.4359 |
| Glass | 0.3175 | 0.3180 | 0.2742 | 0.4862 | 0.2141 | 0.2198 |
| Ionosphere | 0.8776 | 0.8354 | 0.9429 | 0.8955 | 0.9661 | 0.9660 |
| Mammography | 0.3056 | 0.4239 | 0.4295 | 0.4290 | 0.4430 | 0.4007 |
| NSL-KDD | 0.8750 | 0.8885 | 0.7973 | 0.7809 | 0.8301 | 0.8217 |
| Optdigits | 0.2000 | 0.4433 | 0.1274 | 0.0791 | 0.2588 | 0.2732 |
| Pima | 0.6622 | 0.6982 | 0.6912 | 0.7533 | 0.7382 | 0.7672 |
| Pendigits | 0.8293 | 0.8740 | 0.2942 | 0.4656 | 0.7394 | 0.4501 |
| Satellite | 0.8483 | 0.8352 | 0.8211 | 0.7604 | 0.8199 | 0.8270 |
| Satimage-2 | 0.7626 | 0.9746 | 0.9707 | 0.9714 | 0.9774 | 0.9769 |
| Shuttle | 0.9955 | 0.9922 | 0.9612 | 0.9392 | 0.9584 | 0.9517 |
| Thyroid | 0.8284 | 0.7872 | 0.7003 | 0.6819 | 0.5509 | 0.7325 |
| Wbc | 0.7622 | 0.7326 | 0.8144 | 0.8115 | 0.7628 | 0.8075 |
| Wine | 0.9308 | 0.9304 | 0.9313 | 0.9299 | 0.9246 | 0.9313 |
| Average | 0.6913 | 0.7115 | 0.6338 | 0.6554 | 0.6604 | 0.6677 |

Table 18: Detailed Comparison Results of AUC-ROC with different backbones

| Dataset | MLP | MLP MS | Transformer | Transformer MS | Transformer AP | Transformer MS&AP |
|---|---|---|---|---|---|---|
| Arrhythmia | 0.7876 | 0.7732 | 0.7506 | 0.7726 | 0.7826 | 0.7701 |
| Breastw | 0.9977 | 0.9977 | 0.9969 | 0.9972 | 0.9980 | 0.9974 |
| Campaign | 0.8339 | 0.7557 | 0.7528 | 0.8579 | 0.7569 | 0.8125 |
| Cardio | 0.9677 | 0.9677 | 0.9519 | 0.9651 | 0.8945 | 0.9669 |
| Cardiotocography | 0.7400 | 0.7756 | 0.7682 | 0.6796 | 0.7849 | 0.8381 |
| Census | 0.7228 | 0.7249 | 0.6694 | 0.7012 | 0.6196 | 0.4215 |
| Fraud | 0.9139 | 0.9252 | 0.9331 | 0.9272 | 0.9334 | 0.9331 |
| Glass | 0.7216 | 0.7235 | 0.7304 | 0.8373 | 0.6686 | 0.6559 |
| Ionosphere | 0.8514 | 0.8133 | 0.9287 | 0.8736 | 0.9546 | 0.9499 |
| Mammography | 0.8242 | 0.8682 | 0.8240 | 0.8804 | 0.7876 | 0.8791 |
| NSL-KDD | 0.8645 | 0.8697 | 0.7510 | 0.7168 | 0.8113 | 0.7850 |
| Optdigits | 0.8586 | 0.9591 | 0.7718 | 0.6720 | 0.8981 | 0.8615 |
| Pima | 0.6496 | 0.6909 | 0.6791 | 0.7381 | 0.7252 | 0.7353 |
| Pendigits | 0.9929 | 0.9937 | 0.9064 | 0.9514 | 0.9716 | 0.9479 |
| Satellite | 0.7945 | 0.7862 | 0.7650 | 0.6779 | 0.7472 | 0.8063 |
| Satimage-2 | 0.9939 | 0.9984 | 0.9977 | 0.9977 | 0.9970 | 0.9984 |
| Shuttle | 0.9997 | 0.9995 | 0.9959 | 0.9973 | 0.9973 | 0.9978 |
| Thyroid | 0.9809 | 0.9732 | 0.9642 | 0.9595 | 0.9411 | 0.9690 |
| Wbc | 0.9533 | 0.9436 | 0.9471 | 0.9694 | 0.9461 | 0.9627 |
| Wine | 0.9461 | 0.9445 | 0.9476 | 0.9430 | 0.9507 | 0.9476 |
| Average | 0.8697 | 0.8742 | 0.8516 | 0.8558 | 0.8583 | 0.8618 |

# K  DETAILED RESULTS OF DISTANCE MEASUREMENT

We present the detailed performances of AUC-PR and AUC-ROC by utilizing MSE or OT to measure the distance between the basis vector and association prototype in Table 19 and Table 20. Compared to the MSE distance, the OT-based measurement could yield better performance served as the distance metric, demonstrating it is effective for us to formulate both the basis vector and association prototype learning as OT problems and calculate the transport distances between distributions.

Table 19: Detailed Comparison Results of AUC-PR for OT/MSE distance to learn basis vectors (BV) and association prototypes (AP)

| Dataset | MSE-BV MSE-AP | MSE-BV OT-AP | OT-BV MSE-AP | OT-BV OT-AP |
|---|---|---|---|---|
| Arrhythmia | 0.5608 | 0.5618 | 0.6013 | 0.6164 |
| Breastw | 0.9974 | 0.9975 | 0.9973 | 0.9973 |
| Campaign | 0.4424 | 0.3325 | 0.5320 | 0.5826 |
| Cardio | 0.8400 | 0.8391 | 0.8396 | 0.8445 |
| Cardiotocography | 0.5576 | 0.6060 | 0.6722 | 0.6962 |
| Census | 0.1533 | 0.1240 | 0.1478 | 0.2970 |
| Fraud | 0.4881 | 0.3452 | 0.4852 | 0.5377 |
| Glass | 0.2922 | 0.3286 | 0.3685 | 0.3880 |
| Ionosphere | 0.9221 | 0.9812 | 0.9569 | 0.9813 |
| Mammography | 0.3295 | 0.2594 | 0.3290 | 0.4398 |
| NSL-KDD | 0.7981 | 0.8689 | 0.7678 | 0.8823 |
| Optdigits | 0.5621 | 0.5785 | 0.5080 | 0.7957 |
| Pima | 0.7192 | 0.7227 | 0.6724 | 0.7308 |
| Pendigits | 0.2777 | 0.6776 | 0.7355 | 0.9260 |
| Satellite | 0.8474 | 0.8116 | 0.8357 | 0.8433 |
| Satimage-2 | 0.9746 | 0.9195 | 0.9811 | 0.9844 |
| Shuttle | 0.9173 | 0.9341 | 0.9129 | 0.9337 |
| Thyroid | 0.2104 | 0.4379 | 0.2516 | 0.7685 |
| Wbc | 0.7913 | 0.8307 | 0.7718 | 0.8451 |
| Wine | 0.9304 | 0.9304 | 0.9308 | 0.9323 |
| Average | 0.6306 | 0.6543 | 0.6649 | **0.7513** |

Table 20: Detailed Comparison Results of AUC-ROC for OT/MSE distance to learn basis vectors (BV) and association prototypes (AP)

| Dataset | MSE-BV MSE-AP | MSE-BV OT-AP | OT-BV MSE-AP | OT-BV OT-AP |
|---|---|---|---|---|
| Arrhythmia | 0.7537 | 0.7501 | 0.8022 | 0.8147 |
| Breastw | 0.9974 | 0.9974 | 0.9973 | 0.9973 |
| Campaign | 0.7030 | 0.6791 | 0.8078 | 0.8693 |
| Cardio | 0.9628 | 0.9628 | 0.9624 | 0.9653 |
| Cardiotocography | 0.6909 | 0.7290 | 0.8036 | 0.8210 |
| Census | 0.6265 | 0.5427 | 0.6442 | 0.7622 |
| Fraud | 0.9399 | 0.9231 | 0.9391 | 0.9531 |
| Glass | 0.7049 | 0.7588 | 0.7843 | 0.8353 |
| Ionosphere | 0.9061 | 0.9734 | 0.9447 | 0.9738 |
| Mammography | 0.8840 | 0.8093 | 0.8834 | 0.8882 |
| NSL-KDD | 0.6381 | 0.8243 | 0.7205 | 0.8513 |
| Optdigits | 0.9727 | 0.9719 | 0.9602 | 0.9825 |
| Pima | 0.7436 | 0.7177 | 0.6412 | 0.7234 |
| Pendigits | 0.8477 | 0.9701 | 0.9881 | 0.9961 |
| Satellite | 0.8162 | 0.7763 | 0.7852 | 0.7992 |
| Satimage-2 | 0.9983 | 0.9912 | 0.9989 | 0.9995 |
| Shuttle | 0.9960 | 0.9956 | 0.9925 | 0.9965 |
| Thyroid | 0.8114 | 0.9305 | 0.8386 | 0.9750 |
| Wbc | 0.9576 | 0.9668 | 0.9525 | 0.9737 |
| Wine | 0.9445 | 0.9445 | 0.9461 | 0.9507 |
| Average | 0.8448 | 0.8607 | 0.8696 | **0.9064** |

## L  SOFT MASK VISUALIZATION

The purpose of soft masking is to capture intrinsic correlations in normal data by finding which unmasked features can reconstruct the masked features well. In the following, we give a detailed discussion about the soft masking strategy in the raw data space in response to your question:

1) Compared to the regular binary mask with a mask value of either 0 or 1, we apply the soft mask in raw data space with values between 0 and 1, providing a more flexible degree of information blocking and avoiding the complete lost of some features. When applying soft masks, the model can not only choose which features to mask, but also the degree of masking. 2) Note that the relationships across features are relatively regular in each dataset. The data-space masks try to find and automatically learn such regular patterns of relationships across features and embed them into the input. Specifically, some features are more critical for indicating anomalies while others are inconsequential. The soft masks could perform data-adaptive information bias to different features. As shown in Fig. 11 of Appendix L, the soft masks are regular across different features. The data-space masks could uncover the statistical correlations between masked and unmasked positions across data points. 3) The soft masking strategy is not only learnable but also data-related, i.e. assigning different masks for different data, which contributes to finding salient correlations for a specific normal sample. This brings more flexibility to the model and is conducive to learning diverse and optimal masks under which the masked normal data can be reconstructed better than anomalies. 4) Soft masking is more appropriate for tabular data. Compared to the random masking strategy which produces meaningless masks, our learnable masking strategy can not only choose which features to mask but also the degree of masking, generating optimal masks for our purpose. By reconstructing masked positions by capturing their correlations between unmasked positions, we train the soft masks to capture intrinsic correlations existing in the raw data space, and anomalies can be judged by whether deviating from such correlations.

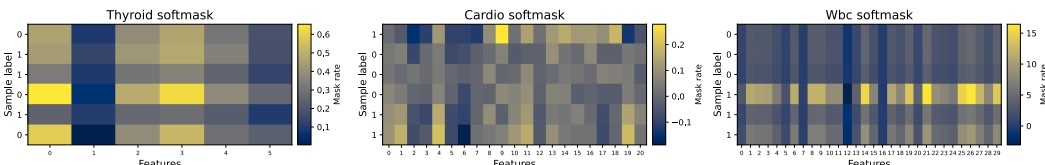

Figure 11: Soft mask visualization

## M  DETAILED RESULTS ON DIFFERENT DATASETS

In Table 21 and Table 22, we listed the detailed results of Figure 2, Figure 3, Figure 4, and Figure 5. It can be seen that our method shows consistently good performances on various datasets.

Table 21: Comparison results of AUC-PR on 20 datasets. The best are bold and the second best are underlined.

| Dataset | KNN | IForest | LOF | OCSVM | GMM | LUNAR | DeepSVDD | GOAD | NeuTralAD | ICL | DTE-C | NPT-AD | MCM | MCM + NPT | Ours |
|---|---|---|---|---|---|---|---|---|---|---|---|---|---|---|---|
| Arrhythmia | 0.3282 | 0.5965 | 0.3134 | 0.3399 | 0.3071 | 0.3602 | 0.5515 | _0.5988_ | 0.5817 | 0.5773 | 0.6609 | 0.4779 | 0.5765 | 0.5956 | **0.6164** |
| Breastw | 0.9600 | 0.9685 | 0.3214 | 0.9582 | 0.9490 | 0.9047 | 0.9024 | 0.9782 | 0.6866 | 0.9459 | 0.9207 | 0.9815 | 0.9902 | **0.9976** | _0.9973_ |
| Campaign | 0.2736 | 0.3300 | 0.2055 | 0.2736 | 0.3139 | 0.2453 | 0.3302 | 0.4518 | 0.3867 | 0.4291 | 0.4877 | 0.4852 | _0.5543_ | 0.4954 | **0.5826** |
| Cardio | 0.3159 | 0.5368 | 0.1914 | 0.4619 | 0.3373 | 0.1733 | 0.7269 | 0.8405 | 0.8570 | 0.8054 | 0.7125 | 0.8216 | _0.8432_ | 0.8352 | **0.8445** |
| Cardiotocography | 0.3387 | 0.4616 | 0.2647 | 0.4101 | 0.3381 | 0.2400 | 0.5976 | 0.6840 | 0.6238 | 0.6443 | 0.5462 | 0.6443 | _0.7007_ | **0.7108** | 0.6962 |
| Census | 0.0920 | 0.0728 | 0.0704 | 0.0848 | 0.0853 | 0.0809 | 0.0756 | 0.1148 | 0.1206 | 0.1850 | 0.1758 | _0.2363_ | 0.2337 | 0.2121 | **0.2970** |
| Fraud | 0.0595 | 0.1396 | 0.0017 | 0.0873 | 0.1088 | 0.0648 | 0.7627 | 0.5076 | 0.4730 | 0.5909 | _0.6343_ | 0.3972 | 0.5884 | **0.6389** | 0.5377 |
| Glass | 0.0764 | 0.0508 | 0.1013 | 0.0421 | 0.0586 | 0.1146 | 0.1637 | 0.1205 | 0.1873 | 0.2296 | **0.5002** | 0.2235 | 0.1752 | 0.2414 | _0.3880_ |
| Ionosphere | 0.9028 | 0.8125 | 0.8256 | 0.8094 | 0.9589 | 0.9511 | 0.8636 | 0.9484 | 0.9818 | 0.9771 | 0.9801 | **0.9875** | 0.9740 | 0.9803 | _0.9813_ |
| Mammography | 0.2088 | 0.2289 | 0.1317 | 0.2213 | 0.1941 | 0.1396 | 0.0429 | 0.1614 | 0.0387 | 0.1792 | 0.4045 | 0.4133 | 0.3173 | **0.4778** | _0.4398_ |
| NSL-KDD | 0.5355 | 0.3787 | 0.5556 | 0.3509 | 0.3685 | 0.5387 | 0.4876 | 0.8536 | 0.8676 | 0.5621 | **0.8932** | 0.8603 | 0.8572 | 0.8237 | _0.8823_ |
| Optdigits | 0.0245 | 0.0669 | 0.0296 | 0.0300 | 0.0317 | 0.0321 | 0.1103 | 0.0847 | 0.1736 | 0.4400 | 0.1465 | 0.1251 | _0.7135_ | 0.3577 | **0.7957** |
| Pima | 0.4893 | 0.4999 | 0.4185 | 0.4341 | 0.4873 | 0.5044 | 0.6409 | 0.6618 | 0.6081 | 0.6462 | 0.6075 | 0.6858 | 0.6759 | _0.7205_ | **0.7308** |
| Pendigits | 0.0702 | 0.4362 | 0.0371 | 0.2279 | 0.0508 | 0.0557 | 0.2161 | 0.3319 | 0.5777 | 0.4003 | 0.4712 | **0.9388** | 0.7338 | 0.7663 | _0.9260_ |
| Satellite | 0.5381 | 0.6164 | 0.4125 | 0.6496 | 0.5092 | 0.4513 | 0.8401 | 0.8077 | _0.8654_ | **0.8976** | 0.8496 | 0.8540 | 0.8502 | 0.8356 | 0.8433 |
| Satimage-2 | 0.2861 | 0.9412 | 0.4125 | 0.9637 | 0.4028 | 0.2742 | 0.7106 | 0.8625 | 0.8367 | 0.8599 | 0.7473 | **0.9859** | 0.9792 | 0.9642 | _0.9844_ |
| Shuttle | 0.1797 | 0.9776 | 0.1293 | 0.8938 | 0.8623 | 0.1816 | **0.9875** | 0.9765 | 0.9804 | _0.9766_ | 0.9430 | 0.9656 | 0.9666 | 0.9165 | 0.9377 |
| Thyroid | 0.3626 | 0.6016 | 0.1482 | 0.3254 | 0.1779 | 0.1474 | 0.5502 | 0.7292 | 0.8095 | 0.6834 | **0.8338** | 0.7851 | _0.8188_ | 0.5385 | 0.7685 |
| Wbc | 0.5641 | 0.6305 | 0.5914 | 0.5439 | 0.4581 | 0.5623 | 0.7535 | 0.7292 | 0.2130 | 0.7795 | 0.6453 | 0.7497 | 0.7466 | _0.7973_ | **0.8451** |
| Wine | 0.2981 | 0.3044 | 0.3637 | 0.1692 | 0.1095 | 0.0585 | 0.9021 | 0.4608 | 0.2425 | 0.3631 | 0.8573 | 0.7635 | _0.9269_ | 0.9260 | **0.9323** |
| Average | 0.3452 | 0.4826 | 0.2763 | 0.4139 | 0.3555 | 0.3040 | 0.5608 | 0.5952 | 0.5556 | 0.6086 | 0.6509 | 0.6691 | _0.7111_ | 0.6916 | **0.7513** |

Table 22: Comparison results of AUC-ROC on 20 datasets. The best are bold and the second best are underlined.

| Dataset | KNN | IForest | LOF | OCSVM | GMM | LUNAR | DeepSVDD | GOAD | NeuTralAD | ICL | DTE-C | NPT-AD | MCM | MCM + NPT | Ours |
|---|---|---|---|---|---|---|---|---|---|---|---|---|---|---|---|
| Arrhythmia | 0.7843 | _0.8615_ | 0.7835 | 0.7978 | 0.7564 | 0.8297 | 0.7502 | 0.8146 | 0.8192 | 0.7937 | **0.8816** | 0.7103 | 0.7826 | 0.8012 | 0.8147 |
| Breastw | 0.9806 | 0.9843 | 0.4761 | 0.9643 | 0.9690 | 0.9711 | 0.7700 | 0.9766 | 0.7987 | 0.9622 | 0.9360 | 0.9834 | 0.9911 | **0.9977** | _0.9973_ |
| Campaign | 0.7447 | 0.7320 | 0.6348 | 0.7368 | 0.7676 | 0.6932 | 0.3951 | 0.7201 | 0.6353 | 0.7032 | 0.7995 | 0.7778 | _0.8619_ | 0.7830 | **0.8693** |
| Cardio | 0.7282 | 0.9247 | 0.6458 | 0.9228 | 0.8703 | 0.5236 | 0.7508 | _0.9639_ | 0.9601 | 0.9140 | 0.8889 | 0.9211 | 0.9635 | 0.9543 | **0.9653** |
| Cardiotocography | 0.5044 | 0.6822 | 0.5217 | 0.6706 | 0.5746 | 0.4932 | 0.6432 | 0.7670 | 0.6799 | 0.6840 | 0.6181 | 0.6840 | 0.8024 | _0.8207_ | **0.8210** |
| Census | 0.6729 | 0.6081 | 0.5716 | 0.6555 | 0.6586 | 0.6410 | 0.5431 | 0.5330 | 0.4986 | 0.6725 | 0.6834 | 0.7008 | 0.7515 | 0.7212 | **0.7622** |
| Fraud | _0.9520_ | 0.9533 | 0.4922 | 0.9562 | 0.9451 | 0.9209 | 0.8846 | 0.9356 | 0.8892 | 0.9143 | 0.9413 | **0.9564** | 0.9025 | 0.8840 | 0.9531 |
| Glass | 0.7840 | 0.6748 | 0.8447 | 0.4927 | 0.5064 | 0.8462 | 0.6375 | 0.6257 | 0.7907 | 0.8196 | **0.9390** | 0.7843 | 0.7480 | 0.7118 | _0.8353_ |
| Ionosphere | 0.9177 | 0.8197 | 0.8562 | 0.8395 | 0.9741 | 0.9642 | 0.9105 | 0.9366 | 0.9776 | 0.9741 | 0.9713 | **0.9805** | 0.9621 | _0.9724_ | 0.9738 |
| Mammography | 0.8510 | 0.8505 | 0.7398 | 0.8741 | 0.8671 | 0.8323 | 0.4807 | 0.4527 | 0.4604 | 0.5653 | 0.8680 | _0.8928_ | 0.8660 | **0.9078** | 0.8882 |
| NSL-KDD | 0.4638 | 0.2289 | 0.5259 | 0.1482 | 0.2046 | 0.4535 | 0.4953 | _0.8524_ | 0.7521 | 0.2665 | 0.8509 | 0.8126 | **0.9606** | 0.6785 | 0.8513 |
| Optdigits | 0.4022 | 0.7469 | 0.4527 | 0.5176 | 0.5478 | 0.4530 | 0.6052 | 0.6936 | 0.7743 | 0.9552 | 0.8254 | 0.8084 | **0.9837** | 0.9537 | _0.9825_ |
| Pima | 0.6913 | 0.6568 | 0.6105 | 0.5904 | 0.6529 | 0.6755 | 0.5861 | 0.6816 | 0.6238 | 0.6231 | 0.6124 | 0.7161 | 0.6503 | **0.7368** | _0.7234_ |
| Pendigits | 0.7539 | 0.9454 | 0.5073 | 0.9259 | 0.7546 | 0.6835 | 0.3064 | 0.9297 | 0.9720 | 0.8334 | 0.9769 | **0.9983** | 0.9906 | 0.9750 | _0.9961_ |
| Satellite | 0.6784 | 0.6868 | 0.5472 | 0.6609 | 0.6394 | 0.6213 | 0.5848 | 0.7356 | _0.8204_ | **0.8805** | 0.7932 | 0.7914 | 0.7949 | 0.7827 | 0.7992 |
| Satimage-2 | 0.9466 | 0.9874 | 0.3026 | 0.9947 | 0.9853 | 0.7068 | 0.9881 | _0.9954_ | 0.9828 | 0.9951 | **0.9995** | 0.9987 | 0.9989 | 0.9989 | **0.9995** |
| Shuttle | 0.6582 | 0.9959 | 0.5289 | 0.9909 | 0.9809 | 0.6331 | 0.9995 | 0.9931 | 0.9950 | 0.9889 | _0.9976_ | 0.9986 | 0.9986 | 0.9940 | 0.9965 |
| Thyroid | 0.9658 | 0.9829 | 0.8385 | 0.9599 | 0.9207 | 0.9476 | 0.9680 | _0.9881_ | 0.9223 | **0.9896** | 0.9762 | 0.9636 | 0.9636 | 0.9310 | 0.9750 |
| Wbc | 0.9392 | 0.9065 | 0.9413 | 0.9408 | 0.9448 | 0.9418 | 0.9340 | 0.9154 | 0.7364 | 0.9087 | _0.9681_ | 0.9577 | 0.9510 | 0.9617 | **0.9737** |
| Wine | 0.6633 | 0.7200 | 0.9367 | 0.5400 | 0.6867 | 0.3800 | 0.9833 | 0.7883 | 0.7383 | 0.7927 | **0.9864** | 0.9567 | 0.9037 | 0.9260 | 0.9507 |
| Average | 0.7541 | 0.7974 | 0.6379 | 0.7590 | 0.7603 | 0.7132 | 0.6957 | 0.8136 | 0.7953 | 0.8079 | 0.8761 | 0.8779 | _0.8914_ | 0.8746 | **0.9064** |

## N  LOSS CONVERGENCE

The convergence trend of MSE loss, orthogonal loss $\mathcal{L}_{orth}$, feature loss $\mathcal{L}_{bv}$, and attention loss $\mathcal{L}_{ap}$ are visualized in Fig. 12, Fig. 13, and Fig. 14. It can be seen these losses could converge and effectively optimize the parameters.

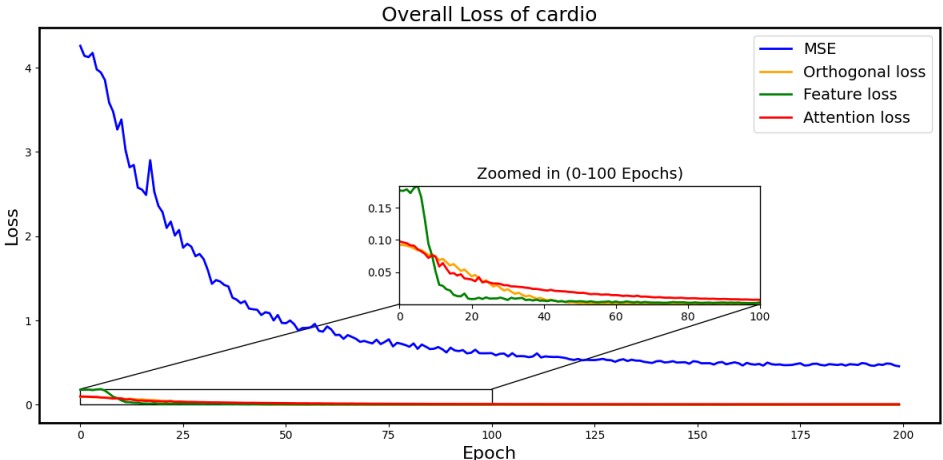

Figure 12: Cardio training loss

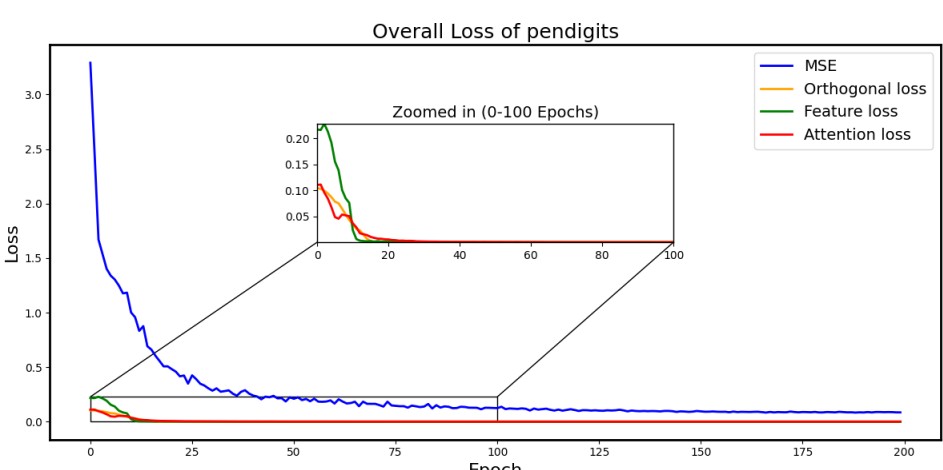

Figure 13: Pendigits training loss

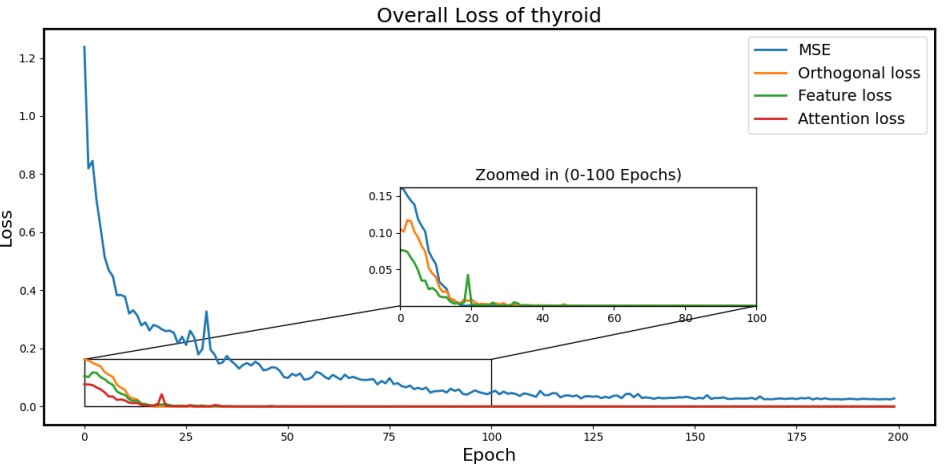

Figure 14: Thyroid training loss

