# OpenReview forum: "PTAD: Prototype-Oriented Tabular Anomaly Detection via Mask Modeling"
_ICLR.cc/2025/Conference — ICLR 2025 Conference Withdrawn Submission_

### Official Review · Reviewer_GRvz · 2024-10-17

**Soundness:** 2
**Presentation:** 2
**Contribution:** 3
**Rating:** 3
**Confidence:** 5

**Summary:**

The authors propose a novel reconstruction-based anomaly detection (AD) method for tabular data that combines previous approaches [4], [2] from the AD literature and the SSL literature [5]. Their approach relies on mask reconstruction as done in [4], [2] in the original data space but is also augmented by three components:
- (i) mask selection in the original data space as done in [4],
- (ii) inter-sample prototype generation in the hidden space to measure distance to the generated prototypes for (1) mask generation in the hidden space during training and (2) anomaly scoring in inference,
- (iii) inter-feature prototypes generation to measure distance to those prototypes for both training and inference.

The main novelty and contribution of the present paper is the use of previously proposed concepts to improve tabular anomaly detection by:
- (1) guiding the learning process towards more desirable equilibria by including new loss functions and generating better suited masks,
- (2) augmenting the vanilla reconstruction-error anomaly score to include more precise measures of inter-feature and inter-sample discrepancies between normal samples and anomalies.

They experiment their method on a benchmark of 20 tabular datasets and show strong performance in comparison to existing methods. Authors conduct extensive ablation studies to provide evidence of the relevance of their approach, and provide explanation on how some key hyperparameters were selected.

**Strengths:**

- The paper is well-structured, and the context of the study is well introduced. Authors detail extensively the existing work found in the relevant literature, identify some weaknesses and propose to address them.
- The present work **augments** the existing mask reconstruction scheme for AD with (i) proper mask selection in inference and during training as done in [4], (ii) prototype-based anomaly scores inspire from prototype generation as done in [5] and shows through their experiments that their methods performs well in comparison with existing methods.
- Their approach is **novel** and combines the work showcased in PTaRL [5] in the SSL for tabular data literature and MCM [4] as well as NPT-AD [2], that all have proven to work well for tabular data.
- The ablation study are extensive and well detailed, demonstrating the relevance of each addition to the existing frameworks.

**Weaknesses:**

- While well-structured the paper can be complicated to follow as style can sometimes be wordy and contain several typos that hinder the overall understandability of the paper. For example, line 235 "*$M_{nh}^k$ denotes the mask value of the $h$-th feature dimension of n-th sample concerning about the k-th basis vector $\beta^{k}$ (...) Furthermore, multiple masks encourage the model to reconstruct samples with various masks, anomalies are prone to be detected by a comprehensive measurement*". Moreover, the term ```nominal``` is sometimes used to refer to ```normal``` samples and this might be confusing, we encourage the authors to stick to the term ```normal```.

- Experiments performed by the authors include only $3$ iterations for $3$ different seeds, which can be considered low in comparison with previous papers that performed experiments 10 or 20 times [1], [2] (while we also acknowledge that [4] also only conducted $3$ runs for each dataset). No statistical tests are mentionned, are any perfomed? Some values are highlighted in the tables while they are unlikely to be significantly different from the second highest value (e.g. Wine dataset, PTAD gets an AUPRC of $0.9323$ vs MCM [4] that gets $0.9269$).

- Presented results are **not reproductible**. We acknowledge that the authors provide the overall code for the main experiments. However, the authors mention a grid search strategy for the learning rate, and do not explicitely mention (i) what objective was used to select the learning rate for this grid search, we assume the loss achieved after 200 epochs?, (ii) the obtained learning rate for each dataset so the experiments can be run again. Only the hyperparameters for the cardio dataset is provided. Moreover, none of the ablation studies can be reproduced without excessive effort while they constitute critical evidence of the relevance of the proposed approach. Overall none of the experiments conducted in section 5 can be reproduced either. **We strongly encourage the authors to provide the full easy-to-run code to support their statements.**

**Questions:**

## Typos:
- Line 150, authors stipulate that [2] introduce NPT while [3] are actually the ones that introduced it, while [2] use it for AD. Moreover, the authors should consider including [2] in the related work section regarding **Tabular Anomaly Detection** to avoid confusion.
- Paragraph **Projection-Space Mask Generation.** in section 4.1. is not easy to follow. In equation (4), the authors write $(z_{nh} - \beta^{k})^2$ and do not define $z_{nh}$. We hypothesize that it corresponds to the $h$-th element of the $H$ dimensional $z_n$ vector, in which case, we do not understand the operation between a scalar and the basis vector of dimension $H$.

## Questions and clarifications
There are a few elements that need clarification:
- **(i)** In Equation (3) in the paper the authors include weights used to construct the masks $\mathbf{W}_i$ for $i\in\{1,2,3\}$ as done in [4]. We suppose that those weights are conjointly learned with the rest of the hyperparameters. Given the overall learning objective, **(i-1) how do you ensure that the learned weights do not yield a trivial solutions, e.g. no features are masked or the masks are identical?** As there is no constraint on the $\hat{\mathbf{X}}$ representation or on the obtained masks, it could very much be that $\mathbf{X} = \hat{\mathbf{X}}$. For example, in [4] the authors include a **diversity loss** that ensures that masks are diverse and non-trivial. [4] mention that,
    "```it raises another vital problem: how to prevent the mask generator generating same and redundant masks. This determines whether MCM can extract different and diverse correlations. Our solution is to constrain the similarity between different masking matrices.```"
**(i-2) What motivated the authors to not include this loss?**

- **(ii)** In the original paper that propose NPTs [3] and in [2] that relies on NPT for AD, both approaches include a mask token that increases the feature representation by one dimension, $(x_i, mask)$, where $x_i$ is a scalar for numerical values and the one-hot representation for categorical features. This representation is then mapped to an $h$-dimensional representation using a linear layer. **(ii-1) How do you perform data-space masking for the categorical features using equation (3) in the paper?** Similarly, is a mask token involved in the pipeline? A detailed description of the pipeline as done in App. C.3. of [3] or App. B of [2] might clarify things. We suggest the authors include such description in the appendix.

- **(iii)** During training, prototypes are obtained from the encoded **masked** representations of normal samples. While constructing the basis vector can be natural for unmasked samples, it appears very complicated to capture normal behaviors in normal samples that may have been masked very differently. Let us consider a very simplistic example where we have a batch of $3$ samples, and that the mask produced in equation (3) in the paper is
    $$
    M^{ds} = \begin{pmatrix} 1 & 0 & 0 \\\\
                             0 & 1 & 0 \\\\
                             0 & 0 & 1
    \end{pmatrix}
    $$
    how can the basis vectors be relevant in that case? More precisely, in the general case, if the mask produced in equation (3) is a diagonal matrix, which is possible since no constraint on the mask in included, **(iii-1) how can the obtained basis vector contain meaningful information?**
    In particular, in the more general case as the number of basis vectors is set to be $5$ in all tested datasets, what if the $M^{ds}$ matrix produces significantly more than $5$ mask types, it seems very unlikely that those basis vector will disentangle properly the representations.
- **(iv)** Prototype generation is also used to capture normal inter-features dependencies. We could be wrong, but it appears to us that aiming at learning inter-feature dependencies on representations that result from a double masking strategy (in the data space and in the encoded space) is particularly challenging and we are surprised that the model could perform such task. **(iv-1) How do you ensure that the inter-feature dependencies learned by the NPT itself does not suffice to discriminate between normal samples and anomalies?**
- **(v)** In inference, since $\mathbf{X}$ contains both normal samples and anomalies, how do you ensure that the basis vector computed in the pipeline still capture normal behavior (both for $s_n^{ap}$ and $s_n^{bv}$). **(v-1) If anomalies originate from a similar distribution (but distinct from the normal distribution), couldn't some obtained basis vectors also capture anomalous behavior thus polluting the anomaly score through anomaly leakage?**
- **(vi)** An average over three runs seems very low. While we acknowledge that [4] also reports an average over $3$ runs, other recent work [1],[2] report an average over $20$ runs which seems more reasonnable. Increasing the number of runs might be relevant, in particular since the training cost seems reasonnable as reported in table 5. Moreover, **(vi-1) could the authors report the standard deviations (as done in [1], [2])?** **(vi-2) Are bold results significantly higher than competing methods?** No mention of a statistical test can be found in the paper.
- **(vii)** Some previous papers [1],[2] have also relied on the F1-Score, it would be best if the authors could include this metrics used in previous benchmarks. In particular, the code provided shows that F1-Score is stored and can be included without too much effort.


Overall, we believe that the present work is promising as it combines existing ideas to provide a novel pipeline for tabular anomaly detection. However given the listed weakness and interrogations, we lean towards reject. However, we are very open to discussion and would be happy to increase our score if the authors are able to clarify our interrogations as well as provide the code to run all the experiments presented in the paper.

[1] Tom Shenkar and Lior Wolf. Anomaly detection for tabular data with internal contrastive learning. In International conference on learning representations, 2022.

[2] Hugo Thimonier, Fabrice Popineau, Arpad Rimmel, and Bich-Liên Doan. Beyond individual input for deep anomaly detection on tabular data. In Forty-first International Conference on Machine Learning, 2024.

[3] Jannik Kossen, Neil Band, Clare Lyle, Aidan N Gomez, Thomas Rainforth, and Yarin Gal. Self-attention between datapoints: Going beyond individual input-output pairs in deep learning. Advances in Neural Information Processing Systems, 2021.

[4] Jiaxin Yin, Yuanyuan Qiao, Zitang Zhou, Xiangchao Wang, and Jie Yang. Mcm: Masked cell modeling for anomaly detection in tabular data. In The Twelfth International Conference on Learning Representations, 2024.

[5] Hangting Ye, Wei Fan, Xiaozhuang Song, Shun Zheng, He Zhao, Dandan Guo, and Yi Chang. Ptarl: Prototype-based tabular representation learning via space calibration. In International Conference on Learning Representations, 2024

---

> ### Author Response · Authors · 2024-11-24
>
> **Response to W3 about the code and hyperparameters**:
> Thanks for your comments. Our learning rate is selected by the convergence speed of training loss, following the same selection criteria as NPT-AD[2]. Furthermore, we have provided the full set of hyperparameters for all datasets in Appendix D. Besides, to facilitate the reproduction of our results, we have provided an anonymized link (https://anonymous.4open.science/r/PTAD-D76D/), where the pre-trained models can be directly downloaded for validation. This allows researchers to verify our results without requiring extensive training efforts.
>
> **Table 8: Datasets hyperparameters. The batch size -1 refers to the input of the entire training set.**
>
> | Dataset          | Epoch | Batch size | Learning rate |
> |------------------|-------|------------|---------------|
> | Arrhythmia       | 200   | -1         | 0.01          |
> | Breastw          | 200   | -1         | 0.01          |
> | Campaign         | 200   | 512        | 0.001         |
> | Cardio           | 200   | 512        | 0.00001       |
> | Cardiotocography | 200   | 512        | 0.001         |
> | Census           | 50    | 32         | 0.01          |
> | Fraud            | 200   | 512        | 0.001         |
> | Glass            | 200   | -1         | 0.0001        |
> | Ionosphere       | 200   | -1         | 0.01          |
> | Mammography      | 200   | 512        | 0.02          |
> | NSL-KDD          | 200   | 512        | 0.01          |
> | Optdigits        | 200   | 512        | 0.01          |
> | Pima             | 200   | -1         | 0.01          |
> | Pendigits        | 200   | 512        | 0.01          |
> | Satellite        | 200   | 512        | 0.01          |
> | Satimage-2       | 200   | 512        | 0.01          |
> | Shuttle          | 200   | 512        | 0.01          |
> | Thyroid          | 200   | 512        | 0.01          |
> | Wbc              | 200   | -1         | 0.00001       |
> | Wine             | 200   | -1         | 0.00001       |
>
> **Response to Question-typos-1 about the reference [2]**:
>
> Thanks for your suggestions. What we originally meant to say in Line 150 ('Thimonier et al. (2024) introduces NPT and showcases its effectiveness and superiority in tabular AD') is in accordance with you. This expression may lead to a misunderstanding. Therefore, we revised the sentence to 'Motivated by NPT, Thimonier et al. (2024) introduce NPT-AD by incorporating both sample-sample and feature-feature dependencies in tabular AD, which showcases its effectiveness and superiority for tabular data.' Additionally, following your suggestion, we have included the work proposed by Thimonier et al. (2024) in the related work section.
>
> **Response to Question-typos-2 about the Projection-Space Mask Generation**:
>
> Thanks. Following your suggestion, we have clarified the paragraph about Projection-Space Mask Generation.
>
> In our formulation, the H-dimensional vector $z_n$ denotes its representation of the-nth sample, where $z_{nh}$ is the $h$-th element of $z_n$. Besides, basis vector $\beta^k$ is also $H$-dimensional and $\beta^k_h$ denotes its $h$-th element. We element-wise compute the distance between $z_n$ and the basis vectors $\beta^k$ in the projection space, which is used to generate the projection-space mask. Specifically, each mask value $M_{nh}^{k}$ is computed by the Euclidean distance between the $h$-th value of the $n$-th representation, i.e., $z_{nh}$, and the $h$-th value of the $k$-th basis vector, i.e., $\beta^k_h$. This distance-based computation helps the model identify relevant patterns in the latent space and determine which features should be masked as suspicious anomalies.

---

### Official Review · Reviewer_d9So · 2024-10-21

**Soundness:** 2
**Presentation:** 2
**Contribution:** 2
**Rating:** 5
**Confidence:** 2

**Summary:**

This paper develops a complex framework for tabular data anomaly detection, consisting of multiple components, such as data-space mask, projection-space mask, and prototype learning.

**Strengths:**

* I am not sure if this is a strength point or not, but I do think the proposed framework is pretty complex.

* The proposed approach can achieve good performance on multiple datasets and outperform various baselines.

* A relatively comprehensive ablation study is also conducted to demonstrate the effectiveness of each component.

**Weaknesses:**

* Although the proposed framework is complex, it seems each component is based on the existing work. It is not a weakness to do the combination. However, it would be better to highlight the unique challenges of the combination and why each component is essential for the whole framework.

* Because the whole framework consists of too many pieces, it makes the whole framework is not easy to follow, and some notations are not very consistent.

    - In Lines 135 -- 154, mixing using $N$ and $n$ is confusing but still guessable. However, based on Equation 1, I cannot get how to derive Z as defined in Line 224.

    - If I understand correctly, the framework should have the basis vectors $\mathcal{B}$ first via the projection space learning and then derive the projection-space mask. If so, it may be better to present the projection space learning first.

    - In the projection space learning part, what does $\theta_E$ indicate? What does $\sigma_{\beta^k}$ in $Q(\mathcal{B})$ indicate?  What is the relationship between $\sigma_{\beta^k}$ and the basis vector $\beta^k$?

   - To be honest, I already felt lost when reading Sec 4.2. Similar questions as above, what does $P(\pi)$ indicate in Line 296? What is the purpose of the objective function defined in Eq 7? What is the expectation for this objective?

**Questions:**

The term "anomaly leakage" is new to me. What is anomaly leakage? In what situation will we observe the issue? Why can the masking strategy solve this issue?

---

> ### Author Response · Authors · 2024-11-24
>
> **Response to W1 about the challenges when introducing our method**:
>
> Thanks for your insightful advice for improving our paper. Firstly, we hope to emphasize that our model is problem-oriented and elaborately designed for tabular data. As the reviewer mentioned, it is challenging to design an appropriate method for tabular AD, which is reflected in the following aspects:
>
> 1) Developing a masking strategy is challenging due to the intricate, heterogeneous, and unstructured nature of tabular data. Deriving commonly utilized random masking may leak a large amount of abnormal information into the reconstruction of tabular data, which is highly unstructured and entangled, leading to irregular reconstruction. Thus, we introduce the data-adaptive masks to find suspicious anomaly locations and reconstruct them with normal information, thus resulting in small or large deviations for instructing normal and anomalies, respectively.
>
> 2) How to select data-adaptive suspicious abnormal masks? The straightforward way is to compare the representations with normal ones. However, the learned representations of existing deep tabular methods are usually entangled and thus cannot support finding the suspicious abnormality. Therefore, we encourage learning the disentangled representation within the projection space. To end this, we introduce a group of learnable orthogonal basis vectors. Furthermore, to capture various data characteristics and learn the diverse inherent relationships, we introduce a multiple mask strategy and only conduct it in the decoder to save computational consumption. Please note that introducing the data-adaptive strategy in not only observation space but also latent space is novel, which is non-trivial.
>
> 3) For compensation to the feature-level patterns as mentioned in 1) and 2), we also investigate the correlation patterns between features to facilitate the modeling of tabular normal patterns and detecting anomalies by introducing a novel association prototype learning. It is beneficial for fitting the heterogeneous and complex characteristics of tabular data.
>
> To sum up, our model aims to improve the tabular AD from two perspectives: one is designing the adaptive mask and the other is learning the correlation patterns across features based on the adaptive masks. It is the interaction and complementarity between these two parts that make our method effective for tabular anomaly detection.
>
> **Respons to W2**:
>
> Sorry for any confusion caused, we have revised our paper to make the descriptions clearer and the logic easier to understand.
>
> a) $N$ is the number of samples, and $n$ is a typo here, revised to $N$. $Z = \Phi_E(\hat{X}; \theta_E) \in \mathbb{R}^{N \times H}$ is the encoded representation of $N$ samples. Here, $\Phi_E$ is the NPT encoder layer composed of an ABD and an ABA layer, which is parameterized by $\theta_E$. $\hat{X}$ is the $N$ masked input obtained from Eq.(3).
>
> b) Thanks for your insightful advice. As mentioned in Weakness-2), we introduce the masks before the projection space learning, as the latter serves as a support for the former, which selects data-adaptive suspicious abnormal masks by learning a disentangled projection space. We have revised the paper to make the logic clearer and easier to understand.
>
> c) $\theta_E$ refers to the parameters of the encoder. $\delta_{\beta^k}$ is the $k^{th}$ basis vectors of the discrete distribution. $\delta_{\beta^k}$ is Dirac function that places a unit point mass at $\beta^k$, and $Q(\mathcal{B}) = \frac{1}{K}\sum_{k=1}^K \delta_{\beta^k}$ means a K-dimensional uniform discrete distribution. This is a commonly used notation in typical OT problems [1][2][3].
>
> d) $P(\pi)$ represents the discrete uniform distribution over $N$ association vectors. The logic of section 4.2 can be stated as follows: i) we first extract the correlations between features in each input data as $\pi_n$, denoted as association vectors. ii) we record the normal correlations as $M$ association prototypes, which need to be learned. iii) The OT-based optimization objective in Eq.(7) is to minimize the distance between the association vectors to the corresponding nearest association prototype. This objective is utilized to learn the association prototypes to record normal features. The intuition behind this is normal patterns tend to approach one of the association prototypes rather than its fusions to alleviate the potential collapse.
>
> Furthermore, for easier understanding of the information flow and logic, we have included our algorithm and training pipeline in Appendix A and Appendix B, respectively.

---

> > ### Comment · Reviewer_d9So · 2024-11-27
> >
> > Thanks for providing the revised manuscript. This version is much much better. I guess the issue is that I have limited knowledge of OT. Therefore, the two key objective functions defined in Equations 5 and 7 still feel hard to follow. Could you use layman's language to explain objective functions?
> >
> > For example, the comment above states, "This objective is utilized to learn the association prototypes to record normal features. The intuition behind this is normal patterns tend to approach one of the association prototypes rather than its fusions to alleviate the potential collapse."
> >
> > My question could be, what does the following statement, "the association prototypes to record normal features", indicate?
> > What is the meaning of "normal patterns tend to approach one of the association prototypes rather than its fusions to alleviate the potential collapse"? What is the connection between OT and the statement here? It might be helpful to provide any examples to illustrate the justification here.
> >
> > I slightly tuned up the score and lowered my confidence, as I am not an expert on OT, which seems critical for the approach.

---

> ### Author Response · Authors · 2024-11-24
>
> **Response to the question about anomaly leakage**:
>
> Thanks for your question. 'Anomaly leakage' is a typical issue in the anomaly detection field [4,5,6]. It has various alternative names or variants, such as 'identical shortcut' and 'identical reconstruction'. Reconstruction-based methods detect anomalies by assuming that a well-trained model (only with normal samples) always produces normal samples regardless of whether the anomalies are within the inputs. In this way, there will be large reconstruction errors for anomalous samples, making them distinguishable from the normal ones. However, with the entire data as input, the reconstruction model can sometimes well reconstruct both the normal and anomalous regions, which appears as returning a good copy of the input disregarding its content. As a result, the anomalous samples can be well recovered with the learned model and hence become hard to detect. This will lead to lower anomaly scores in abnormal regions and thus failure of anomaly detection. Especially when encountering complex data distribution, such as heterogeneous unstructured tabular data, reconstruction methods tend to suffer from this 'Anomaly leakage' problem. It requires the model to work extremely hard to learn the complex data distribution. From this perspective, learning an “identical shortcut” appears as a far easier solution. Moreover, the model trained by this kind of data is more likely to be overfitting. Introducing our masking strategy could block the anomalous information from the origin and prevent the model from reconstructing anomalous, as those suspicious anomalous regions are already masked. To this end, the anomalous samples cannot be well recovered with the normal information and hence become easier to detect.
>
> [1] Gabriel Peyr´e, Marco Cuturi, et al. Computational optimal transport: With applications to data science. Foundations and Trends® in Machine Learning, 11(5-6):355–607, 2019.
>
> [2] Pavel Dvurechensky, Alexander Gasnikov, and Alexey Kroshnin. Computational optimal transport: Complexity by accelerated gradient descent is better than by sinkhorn’s algorithm. In International conference on machine learning, pp. 1367–1376. PMLR, 2018.
>
> [3] Lenaic Chizat, Gabriel Peyr´e, Bernhard Schmitzer, and Franc¸ois-Xavier Vialard. Scaling algorithms for unbalanced optimal transport problems. Mathematics of Computation, 87(314):2563–2609, 2018.
>
> [4] Perera, P.; Nallapati, R.; and Xiang, B. 2019. OCGAN: one- class novelty detection using gans with constrained latent representations, in CVPR.
>
> [5] Zhiyuan You, Lei Cui, Yujun Shen, Kai Yang, Xin Lu, Yu Zheng, and Xinyi Le. A unified model for multi-class anomaly detection. In S. Koyejo, S. Mohamed, A. Agarwal, D. Belgrave, K. Cho, and A. Oh (eds.), Advances in Neural Information Processing Systems, volume 35, pp. 4571–4584. Curran Associates, Inc., 2022.
>
> [6] Yao, Xincheng, et al. One-for-all: Proposal masked cross-class anomaly detection. Proceedings of the AAAI Conference on Artificial Intelligence. Vol. 37. No. 4. 2023.

---

> > ### Comment · Reviewer_d9So · 2024-11-27
> >
> > Thank you for sharing the references.

---

### Official Review · Reviewer_JPNG · 2024-10-26

**Soundness:** 3
**Presentation:** 3
**Contribution:** 1
**Rating:** 3
**Confidence:** 4

**Summary:**

The PTAD (Prototype-Oriented Tabular Anomaly Detection) model introduces advanced mask modeling and prototype learning techniques to enhance anomaly detection in tabular data. To address the issue of anomaly leakage, PTAD applies mask modeling in both data and projection spaces. A soft masking strategy selectively conceals potentially abnormal regions in the data space. Simultaneously, in a disentangled projection space, PTAD creates learnable masks based on orthogonal basis vectors, emphasizing feature-wise discrepancies that aid in obscuring suspicious areas while maintaining nominal patterns for accurate reconstruction. PTAD’s prototype learning further strengthens its anomaly detection by capturing global feature dependencies in tabular data. These “association prototypes” represent typical correlations among nominal features and are learned as part of the model's decoding process. This is framed as an optimal transport (OT) problem, aligning the reconstructed data distribution with the prototype distribution to emphasize nominal patterns and assist in accurate anomaly scoring. Both the projection-space mask modeling and association prototype learning are formulated as OT problems, which calibrate the anomaly scoring based on the degree of deviation from normal patterns. The model’s multi-branch decoder allows for parallel reconstructions of masked representations, capturing complex feature relationships. This integration of data-adaptive masks, OT-aligned prototype learning, and a multi-branch decoding approach allows PTAD to capture nominal feature dependencies effectively, resulting in robust and interpretable anomaly detection across a range of tabular benchmarks.

**Strengths:**

- The manuscript is well-written and easy to follow. The maths is clearly set out, the individual parts justified and the influence demonstrated.
- The paper offers a new perspective and a novel optimization strategy for the application of mask modeling in combination with transformers in the field of anomaly detection.
- The proposed approach performs best in the existing setup compared to the chosen baselines.

**Weaknesses:**

- The criteria for the selection of the data sets selected for the evaluation are not set out (see questions).
- Although current baselines were used, baselines that perform strongly in other works were omitted (see questions).
- Due to the first two points and the combination of both, there is a lack of comparability to be able to make a definitive assessment of the method.

**Questions:**

Q1: The ODDS used contains over 30 pure table datasets, of which only 12 are used; ADBench contains 57 datasets, not just eight as used in this work (there are partial overlaps between the data sets, but this does not change the main problem). How were the data sets selected, why were the others not used and what is the performance of the other data sets? (Answering these questions could lead to further questions).

Q2: Why was a separate benchmark created instead of following previous work (which were used as baseline methods and are therefore known and could probably have been used directly) such as [1] and [2], which all use more data sets or well-known benchmarks?

Q3: Some models from the PyOD library were used as baselines. How were these models selected? Why were methods such as LUNAR, KPCA and GMM, which show strong results in papers such as [3], not used?


Despite the apparent weaknesses regarding evaluation and comparability, this work is well-written and interesting. I would put the marginal acceptance for now, but I will be happy to raise my score if my questions are answered appropriately and the results continue to be sustainable.


[1] Victor Livernoche, Vineet Jain, Yashar Hezaveh, and Siamak Ravanbakhsh. On diffusion modeling  for anomaly detection. In The Twelfth International Conference on Learning Representations, 2024. URL https://openreview.net/forum?id=lR3rk7ysXz.

[2] Hugo Thimonier, Fabrice Popineau, Arpad Rimmel, and Bich-Lien Doan. Beyond individual input for deep anomaly detection on tabular data. arXiv preprint arXiv:2305.15121, 2023.

[3] Roel Bouman, Zaharah Bukhsh, and Tom Heskes. Unsupervised anomaly detection algorithms on real-world data: how many do we need? Journal of Machine Learning Research, 25(105):1–34, 2024

---

### Official Review · Reviewer_vtQS · 2024-10-29

**Soundness:** 2
**Presentation:** 2
**Contribution:** 2
**Rating:** 6
**Confidence:** 2

**Summary:**

This paper aims to tackle two key challenges in tabular anomaly detection: representation entanglement and the lack of global information. It proposes a new framework that combines mask modeling with prototype learning, leveraging optimal transport to align data-adaptive masks with association prototypes. Extensive experiments on 20 tabular benchmarks demonstrate the effectiveness of the proposed approach.

**Strengths:**

1. The paper proposed a new framework for tabular anomaly detection.

2. The experiments, including comprehensive ablation studies, are thorough and effectively demonstrate the method’s performance.

**Weaknesses:**

1. I was wondering about the role of conducting masking strategy in the raw data space. Given that common tabular data features are unordered and lack the natural sequential or contextual relationships found in images or text, how does the masking strategy effectively capture meaningful correlations in tabular data?

2. The writing and some notations could be improved for clarity. As I am not an expert in anomaly detection, I found certain sections, particularly the motivations behind key design choices, challenging to follow.

**Questions:**

1. In line 139, what does the dimension $e$ represent, and how is $H^0$ constructed? Additionally, in Eq. (1), what is the dimension $h$, and why does stacking on axis $n$ yield a tensor with dimensions $N \times d \times e$? Clarifying these aspects would help in understanding the operations in Non-Parametric Transformers.

2. What specific advantage does soft masking offer in the data space?

---

### Note · Authors · 2025-01-14

**Comment:**

Thanks for all your efforts in reviewing our paper.

**Withdrawal Confirmation:**

I have read and agree with the venue's withdrawal policy on behalf of myself and my co-authors.